# 32-year record-high surface melt in 2019/2020 on the northern George VI Ice Shelf, Antarctic Peninsula

Alison F Banwell[1,2], Rajashree Tri Datta[3,4,5], Rebecca L Dell[2], Mahsa Moussavi[6,1], Ludovic Brucker[3,7], Ghislain Picard[8], Christopher A Shuman[3,9] Laura A Stevens[10,11]

1) Cooperative Institute for Research in Environmental Sciences (CIRES), University of Colorado, Boulder, CO, USA
2) Scott Polar Research Institute (SPRI), University of Cambridge, Cambridge, UK
3) Cryospheric Sciences Laboratory, NASA Goddard Space Flight Center, Greenbelt, MD, USA
4) Earth System Science Interdisciplinary Center, University of Maryland, College Park, MD, USA
5) Department of Atmospheric and Oceanic Sciences (ATOC), University of Colorado Boulder, USA
6) National Snow and Ice Data Center (NSIDC), University of Colorado Boulder, CO, USA
7) Goddard Earth Sciences Technology and Research Studies and Investigations, Universities Space Research Association, Columbia, MD, USA
8) Univ. Grenoble Alpes, CNRS, Institut des Géosciences de l'Environnement (IGE), UMR 5001, 38041 Grenoble, France
9) Joint Center for Earth Systems Technology, University of Maryland, Baltimore County, Greenbelt, MD, USA
10) Department of Earth Sciences, University of Oxford, Oxford, UK
11) Lamont-Doherty Earth Observatory of Columbia University, Palisades, NY, USA

*Correspondence to*: Alison F Banwell (alison.banwell@colorado.edu)

**Abstract** In the 2019/2020 austral summer, the surface melt duration and extent on the northern George VI Ice Shelf (GVIIS) was exceptional compared to the 31 previous summers of distinctly lower melt. This finding is based on analysis of near-continuous 41-year satellite microwave radiometer and scatterometer data, which are sensitive to meltwater on the ice-shelf surface and in the near-surface snow. Using optical satellite imagery from Landsat 8 (since 2013) and Sentinel-2 (since 2017), record volumes of surface meltwater ponding were also observed on the northern GVIIS in 2019/2020, with 23% of the surface area covered by 0.62 km$^3$ of ponded meltwater on 19 January. These exceptional melt and surface ponding conditions in 2019/2020 were driven by sustained air temperatures $\geq 0$°C for anomalously long periods (55–90 hours) from late November onwards, which would have limited meltwater refreezing. The sustained warm periods were likely driven by warm, low-speed ($\leq 7.5$ ms$^{-1}$), northwesterly and northeasterly winds, and not by foehn wind conditions, which were only present for 9 hours total in the 2019/2020 melt season. Increased surface ponding on ice shelves may threaten their stability through increased potential for hydrofracture initiation; a risk that may increase due to firn air content depletion in response to near-surface melting.

## 1. Introduction

Since the 1950s, the Antarctic Peninsula (AP) (Fig. 1a) has experienced faster increases in ocean and atmospheric warming than the rest of the Antarctic Ice Sheet (Siegert et al. 2019; Smith et al 2020; Trusel et al. 2015). The rate of mass loss from the AP has tripled since the 1990s, with an average of 24 Gt yr$^{-1}$ from 1979 to 2017, and an acceleration of 16 Gt yr$^{-1}$ per

decade (Rignot et al., 2019). Mass loss is currently focused at marine margins, where the mass balance is controlled by complex interactions between the ice, ocean, atmosphere and inland bed conditions (Scambos et al., 2000; Bell et al., 2018; Shepherd et al., 2018; Tuckett et al. 2019; Smith et al., 2020). An important part of this system are ice shelves, which have a total area of ~120,000 km$^2$ around the AP (Siegert et al., 2019), and act to buttress the inland grounded ice flowing into the ocean (Scambos et al. 2004; De Rydt et al., 2015; Fürst et al., 2016; Gudmundsson et al., 2019).

Ice-shelf surface melting, which results in surface lowering, and if sustained, thinning (Paolo et al., 2015), is connected to ice-shelf stability as follows. In warm summers, meltwater produced at the ice-shelf surface is stored in the perennial snowpack ('firn'). Refreezing of this meltwater releases latent heat into the firn, causing additional melting, firn saturation, and firn air content depletion; eventually facilitating meltwater to pond on the ice-shelf surface (Holland et al., 2011; Kuipers Munneke et al., 2014). Extensive surface ponding (Kingslake et al., 2017; Arthur et al., 2020a; Dell et al., 2020) may threaten ice-shelf

stability due to stress variations associated with overall meltwater movement, ponding and drainage (Scambos et al., 2000, 2003; MacAyeal et al., 2003; Banwell and MacAyeal, 2015; Banwell, 2017; Banwell et al., 2019). These processes may initiate meltwater-induced vertical fracturing ('hydrofracturing') (Van der Veen, 2007; Dunmire et al., 2020; Lai et al. 2020), especially if the ice shelf is already damaged with a high density of crevasses (Lhermitte et al., 2020). The near-complete collapse of the Larsen B Ice Shelf in 2002 is arguably the most famous break-up event due to its rapidity and extent (e.g.

Scambos et al., 2013) and may have been driven by the drainage of ~3000 lakes (Banwell et al., 2013, Robel and Banwell, 2019; Leeson at al., 2020). However, surface melting has also been implicated in the large-scale collapse events of the Prince Gustav and Larsen A ice shelves over just a few days in late January 1995 (Rott et al., 1996; Doake et al., 1998; Scambos et al., 2003; Glasser et al., 2011), and in other, smaller-scale collapses of the Wilkins, Larsen B, George VI, and Larsen A ice shelves (Scambos et al. 2003, 2009; Cook and Vaughan, 2010).

Occurrences of extreme melt seasons can lead to substantial changes that may potentially impact the mass balance of the AP and consequently global sea level rise. In the austral summer of 2019/2020, widespread surface meltwater ponding was observed on ice shelves, low-elevation outlet glaciers and on ice-capped islands of the AP (Fig. 1a). Out of all AP ice shelves, the most extensive area of surface meltwater ponding in 2019/2020 was observed on the northern George VI Ice Shelf (GVIIS); the focus of this study (Fig. 1b). However, as Fig. 1a shows, in 2019/2020, surface meltwater ponding was also prevalent on

the northwestern Larsen C (Bevan et al., 2020), the eastern Wilkins (also visible in the bottom left corner of Fig 1b), and the northern and northwestern Bach ice shelves. This extensive surface ponding across the AP and was accompanied by a record-high (as of yet unverified) instantaneous surface air temperature of 18.4°C, recorded by an automatic weather station (AWS) at Esperanza on the northern tip of the AP on 6 February 2020 (https://public.wmo.int/en/media/news/new-record-antarctic-continent-reported, last access: 8 January 2021).

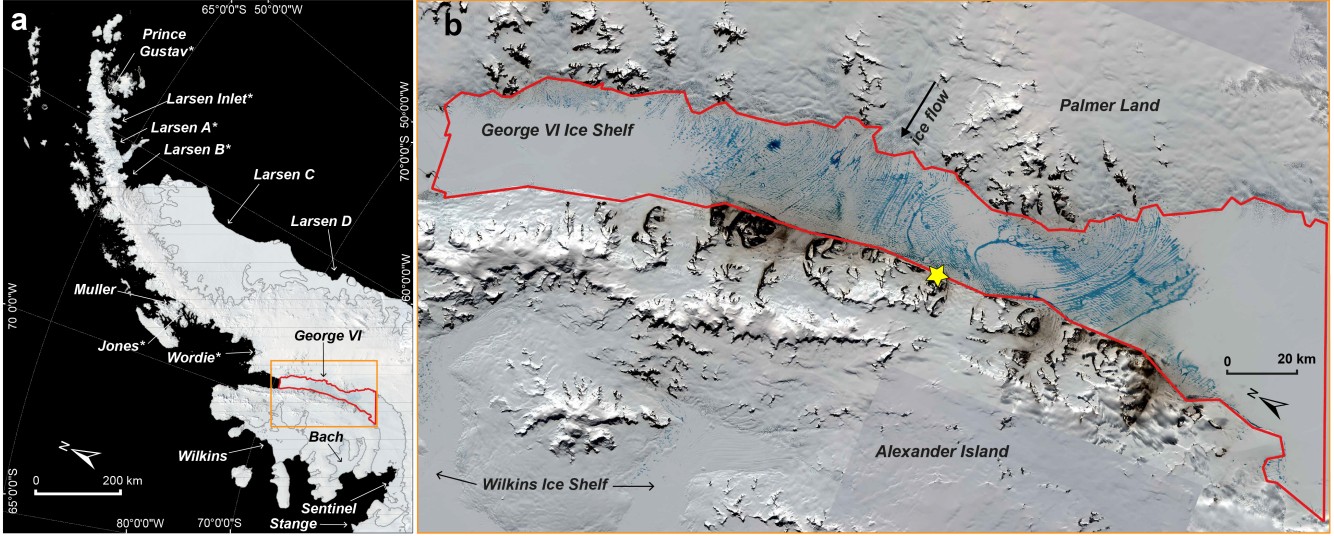


**Figure 1**. (**a**) Mosaic of cloud-free Moderate Resolution Imaging Spectroradiometer (MODIS) images over the AP from 19 January to 7 February 2020. The MODIS mosaic is sea ice-masked and the ice shelves are delineated with grey lines using the U.S. National Ice Center Operational Antarctic Ice Front and Coastline Data Set 2017-2020 (Readinger, 2020). Ice shelves are labelled with white text; those with * have lost > 50% of their original area since the 1950s (Cook and Vaughan, 2010). The

red outline shows the study's Area of Interest (AOI) over north GVIIS. The orange box depicts the area shown in (b). (**b**) A mosaic of optical images over the northern GVIIS AOI. All images are Sentinel-2 tiles dated 19 January 2020, apart from the two darker tiles (top right and lower right, outside of the AOI) which are Landsat 8 image tiles from 17 and 19 January 2020. The study AOI is delineated by the red outline, and the yellow star shows the location of the Fossil Bluff AWS.

**2. Study Site**

GVIIS is located in the southwest AP between Alexander Island and Palmer Land (Fig. 1). With an area of ~23,500 km$^2$ (Rignot et al., 2013), it is the second largest remaining ice shelf on the AP after the Larsen C. GVIIS has two ice fronts, separated by ~450 km along its centreline: a northern ice front that calves into Marguerite Bay, and a southern ice front that terminates into the Ronne Entrance (Holt et al., 2013). GVIIS is structurally complex, with distinct flow units originating in

Palmer Land flowing across to, and impinging against, Alexander Island (Reynolds and Hambrey, 1988; Hambrey et al., 2015; Davies et al., 2017), resulting in a dominantly compressive flow regime (LaBarbera and MacAyeal, 2011). The ice shelf decelerates as it flows westwards across the sound, with ice velocities on the northern GVIIS varying from ~ 400 m yr$^{-1}$ near the grounding line to ~ 30 m yr$^{-1}$ near Alexander Island (Bishop and Walton 1981). This complex flow regime controls ice-shelf thickness, which varies from ~100 m at both ice fronts to ~600 m in the centre (Smith et al., 2007; Davies et al. 2017).

Compared to the southern GVIIS (~72°00' S to ~77°00' S), the northern GVIIS (~70°30' S to ~72°00' S) experiences higher surface summer melt rates (< 250 mm w.e. yr$^{-1}$; Trusel et al., 2013; Datta et al., 2018) and lower accumulation rates (< 200 kg

m$^{-2}$ yr$^{-1}$; Bishop and Walton, 1981; Reynolds, 1981); the latter is attributed to the presence of a precipitation shadow down-wind of Alexander Island (Bishop and Walton, 1981). As a result, winter snowfall on the northern GVIIS rarely lasts through the summer (Holt et al., 2013) and extensive areas of ponded surface water have been observed here since at least the early 1940s (Wager, 1972; Reynolds, 1981). However, as these surface lakes have generally been observed to refreeze at the end of each austral summer, with only limited evidence of meltwater drainage into ice-marginal moulins (Reynolds, 1981), minimal mass is lost through surface melting. Instead, mass is mostly lost due to high basal melt rates of < 6 m yr$^{-1}$ (Adusumilli et al., 2020), attributed to the warm Circumpolar Deep Water current that extends under the entire length of the GVIIS (Holland et al., 2010; Pritchard et al., 2012), though rates of basal melting are greatest at the ice shelf's southern end (Adusumilli et al., 2020; Smith et al. 2020). High basal melt rates have resulted in sustained thinning rates of < 2 m yr$^{-1}$ for the southern GVIIS (Pritchard et al., 2012), which together with frontal calving (Pearson and Rose, 1983; Reynolds and Hambrey, 1988; Lucchitta and Rosanova, 1998), have contributed to the ice shelf's negative net mass balance since at least 2003 (Rignot et al. 2013; Paolo et al., 2015). As an example, Rignot et al. (2019) estimated that the GVIIS lost 9 Gt of mass in 2017, compared to a balance flux of 70 ± 4 Gt yr$^{-1}$. Due to the strong buttressing forces that the GVIIS provides relative to the large volume of grounded ice in Palmer Land, if this ice shelf were to completely collapse, the resultant acceleration of the inland glaciers would add < 8 mm to global sea levels by 2100, and < 22 mm by 2300 (Schannwell et al., 2018). In contrast, Schannwell et al. (2018) calculate that sea level contributions resulting from the collapse of the much larger Larsen C Ice Shelf would be relatively low (< 2.5 mm by 2100, < 4.2 mm by 2300).

On the northern GVIIS, three types of surface lake patterns usually form each summer. The principal pattern of lakes, which are generally the most extensive in area, is aligned with the ice flowlines (Reynolds, 1981; Smith et al., 2007); similar to the dominant pattern of lakes on the Amery Ice Shelf (Hambrey and Dowdeswell, 1994). This set of lakes is intersected by a second pattern of generally smaller, ribbon-type lakes, which lie parallel to the prevailing wind (Reynolds, 1981), suggesting that wind processes initiate the surface depressions that meltwater then fills. These first two sets of lakes appear to remain in similar locations each year due to the ice shelf's overall compressive flow, i.e. unlike the situation on most ice shelves where lakes move with ice flow towards the shelf front (Banwell et al., 2014; Langley et al., 2016; Arthur et al., 2020b). The third set of lakes are the deepest and exist within pressure ridge complexes along the western margin of the ice shelf, onto which ice-shelf flow is directed (Reynolds, 1981). These lakes are therefore en échelon (i.e. closely spaced, sub-parallel) in shape and propagate along the ice-shelf margin, hence have been referred to as 'travelling lakes' (LaBarbera and MacAyeal, 2011).

Unlike other AP ice shelves that have fully or partially disintegrated due to high rates of surface and/or basal melting, the retreat of GVIIS thus far has been relatively gradual, despite this ice shelf having the most extensive meltwater ponding and the longest history of surface lakes of any AP ice shelf (Smith et al., 2007). This is likely due to the GVIIS' unique geographical setting with its dominantly compressive flow regime, as described above, enabling it to support a large volume of surface meltwater (Alley et al., 2018; Lai et al., 2020).

In this study we focus on the northern area of the GVIIS only; defined as our Area Of Interest (AOI) (see Fig. 1b, location shown by the red outline) with a total area of 7850 km$^2$. This is the region where a high density of surface lakes are often observed each melt season.

## 3. Data and Methods

To quantify our understanding of surface melt over the northern GVIIS for the austral summers from 1979/1980 to 2019/2020, we analyse large-scale melt information from 25-km gridded passive microwave observations for both the AP and the northern GVIIS. For the northern GVIIS, these data are corroborated by smaller-scale (4.45-km) active microwave observations available from 2007/2008 to 2019/2020. For austral summers from 2013/2014 to 2019/2020, we also calculate volumes of ponded meltwater on the northern GVIIS from all available cloud-free optical images from the Landsat 8 (2013 to 2020) and Sentinel-2 (2017 to 2020) satellites. Both our microwave-derived melt and optical image-derived surface ponding results are evaluated alongside surface air temperature and wind data from the British Antarctic Survey (BAS) Fossil Bluff AWS (1979 to 2020) on the northwest margin of the GVIIS (Fig. 1b, yellow star).

### 3.1 Large-scale microwave radiometer observations of melt

Microwave radiometer observations of melt, expressed as brightness temperatures, depend primarily on the snow temperature profile and emissivity (Zwally, 1977). When liquid water exists in the snow, there is a significant increase in the absorption, and therefore an increase in the microwave emissivity, resulting in a higher brightness temperature. Large-scale melt information over the AP, including the GVIIS, was derived from microwave radiometer (i.e. passive) observations using the 1979 to 2020 near-daily 25-km melt product (version 2) of Picard et al. (2007) and Picard and Fily (2006), distributed on a polar stereographic grid. This melt/no-melt product, which has been used in several previous studies (e.g., Magand et al., 2008; Brucker et al., 2010; Wille et al., 2019), is based on the algorithm of Torinesi et al. (2003) that identifies the higher microwave brightness temperatures corresponding to melt using the radiation observed at 19 GHz in horizontal polarization. If the observed brightness temperature on a given day exceeds an empirical threshold (defined by the mean and variability of the brightness temperatures observed during the previous winter season, when melt did not occur), the algorithm reports melt in the 25-km grid cell. Throughout this paper, we use the word 'melt' when referring to the presence of liquid meltwater (either in the near-surface snow, or on the surface) in the microwave data, but note that we are not referring to the process of active melting; information that specific cannot be obtained from passive microwave data.

The 1979 to 2020 brightness temperature time series was acquired by five successive sensors. The Scanning Multichannel Microwave Radiometer (SMMR), on the Nimbus 7 satellite launched in late October 1978, collected data at 18 GHz (while the sensor operated every other day, daily-averaged brightness temperatures were used as input). Starting in 1987, the series of Special Sensor Microwave Imager (SSMI) sensors on the Department of Defense Meteorological Satellite Program (DMSP) platforms F8, F11, F13 and F17 collected data at 19 GHz. Of note, there was a significant data gap between 3 December 1987 and 14 January 1988, and therefore we do not include any data from this melt season in our analysis. Although the melt data

are provided with a spatial resolution of 25-km, the radiometers' 3-dB field of views at 19 GHz are far larger (e.g., 69 km x 43 km for SSMI). Grid cells with surface elevations > 1700 m a.s.l. were masked out so that melt over the ice shelves was predominantly analysed and so that large topographic features (i.e., mountain peaks) in the radiometers field-of-view were avoided.

Based on radiative transfer simulations, radiometer brightness temperatures at 19 GHz are typically sensitive to melt down to a snow depth of ~2 m (Picard et al., 2007; Leduc-Leballeur et al., 2020).  Wet snow has a very high emissivity compared to dry snow, but a flat surface of liquid water has a low emissivity as well (Zwally, 1977). Therefore, at the transition from dry to wet snow, brightness temperatures increase quickly (i.e. indicating the presence of meltwater), but if melt intensifies, resulting in the formation of surface lakes, the brightness temperatures decrease. This effect has been observed over sea ice

when melt ponds are extensive (e.g., Kern et al., 2020). Cautious interpretation of the "melt day" maps is therefore required, particularly if surface ponding represents a large proportion of a grid cell's total area.

For austral melt seasons from 1979/1980 to 2019/2020 (apart from 1987/1988 due to its missing data), and for each 25-km grid cell, we calculated the daily time series of microwave radiometer-derived melt/no melt, and the cumulative melt days each melt season (defined as 1 November to 31 March inclusive). This was done for both the whole AP (i.e. extent of Fig. 1a) and

165 for the northern GVIIS AOI (Fig. 1b, red outline).

### 3.2 Small-scale microwave scatterometer observations of melt

For the northern GVIIS, we also derive smaller-scale melt information from an enhanced resolution C-band (5.225 GHz) VV polarization radar backscatter image time series collected by EUMETSAT's Advanced SCATterometer (ASCAT), aboard the tandem polar-orbiting satellites MetOp-A and MetOp–B. The 4.45-km enhanced product is obtained by applying the

170 Scatterometer Image Reconstruction (SIR) algorithm with filtering (Lindsley and Long, 2016), which is used to improve the spatial resolution of irregularly and oversampled data (Early and Long, 2001). The effective spatial resolution was estimated at ~12-15 km; three-fold finer than the effective resolution of the SMMR/SSMI-based product (~50-km). For each day, and for each grid cell, melt is assumed to be present when the ASCAT signal is lower than the winter mean signal minus 3 dB, as proposed by Ashcraft and Long (2006) using a melt model and QuikSCAT Ku-band (13.4 GHz) observations.  Where snow

and firn layers are completely frozen, the C-band penetration depth is of the order of meters to tens of metres, but where snow and firn layers have a high volumetric fractions of meltwater, the penetration depth is likely to be up to 10s of centimetres only (Weber Hoen and Zebker, 2000). As the penetration depth at 5 GHz in dry snow/firn is larger than at 19 GHz, ASCAT C-band radar is likely to be more sensitive to melt at depth than microwave radiometers at 19 GHz.

For austral melt seasons from 2007 to 2020, we calculate scatterometer-derived cumulative melt days for each 4.45-km grid

cell over our study AOI.

### 3.3 Landsat 8 and Sentinel-2 derived meltwater areas and volumes

To calculate the time series of areal extents, depths, and therefore total volumes of surface meltwater lakes on the northern GVIIS for the seven austral summers from 2013/2014 to 2019/2020, we applied the threshold-based algorithm developed by Moussavi et al. (2020) to selected multispectral imagery (see below) from Landsat 8 (30 m resolution, since 2013) and Sentinel-2 (10 m resolution, since 2017). Technical specifications for Landsat 8's Operational Land Imager data are available online from NASA (https://landsat.gsfc.nasa.gov/operational-land-imager-oli/), and technical specifications for Sentinel-2's MultiSpectral Instrument, are available online from ESA (https://sentinels.copernicus.eu/web/sentinel/technical-guides/sentinel-2-msi). Analysis of pre-2013 optical imagery could have been undertaken by tuning Moussavi et al's (2020) threshold-based algorithm for the Landsat 7 Enhanced Thematic Mapper Plus (ETM+) sensor. However, significant data are missing since May 2003 due to the failure of the Scan Line Corrector (SLC) on ETM+ causing SLC-off gaps. Therefore, lake volumes derived from this sensor would not be easily comparable to Landsat 8 and Sentinel-2, and therefore would not necessarily extend the time series.

Moussavi et al's (2020) method, developed in parallel for Landsat 8 and Sentinel-2, combines separate threshold-based algorithms to detect (1) lakes, (2) rocks, and (3) clouds. Optimal thresholds for each band and band combination (e.g. Normalized Difference Water Index (NDWI), Normalized Difference Snow Index (NDSI) and others) were determined by creating a training dataset based on selected Landsat 8 and Sentinel-2 images, which represented spectral properties of several classes (e.g. lakes, slush, snow, clouds, rocks, cloud shadows). Most notably, to classify liquid water-covered pixels, the NDWI is used (Pope et al., 2016; Bell et al., 2017) with NDWI thresholds of > 0.19 and > 0.18 for Landsat 8 and Sentinel-2, respectively. Subsequently, to calculate the water depths of those pixels determined to be water-covered, Moussavi et al. (2020) apply a physically-based algorithm that has more commonly been applied in Greenland (Sneed and Hamilton, 2007; Banwell et al., 2014; Pope et al., 2016; Williamson et al., 2018), and more recently, Antarctica (Bell et al., 2017; Dell et al., 2020; Arthur et al., 2020b). This algorithm calculates lake water depth using the rate that sunlight passing through a water column is attenuated with depth, lake-bottom albedo, and optically deep water reflectance (Philpot, 1989). This approach makes a number of assumptions, including that: 1) the lake bottom has a homogenous albedo; 2) there is little to no particulate matter in the water column to alter its optical properties; and 3) there is minimal wind-induced surface roughness (Sneed and Hamilton, 2007).

All Landsat 8 and Sentinel-2 images acquired from 1 November to 31 March each austral summer with a solar angle of > 15 degrees, with $\geq 0.45$ km$^2$ water-covered pixels (equivalent to 500 Landsat pixels, or 4500 Sentinel-2 pixels; Moussavi et al., 2020), and which overlapped our study's AOI (Fig. 1b, red outline), were analysed using the methods described above. Once the images had been analysed, all tiles with the same date were mosaiced together and then clipped to a mask of our AOI in the Geographic Information System package, QGIS v3.2. In total, for Landsat 8, we analysed mosaiced images for 191 dates from 6 December 2013 to 12 March 2020, and for Sentinel-2, we analysed mosaiced images for 14 dates from 3 January 2017 to 19 January 2020. Of those images, nine Landsat 8 and Sentinel-2 image mosaics had the same dates, so we merged those; first, by resampling the Sentinel-2 data (10 m resolution) to the resolution of Landsat (30 m), and second, by keeping

overlapping water-covered pixels in preference to dry pixels, and by keeping the largest depths of the overlapping water-covered pixels. This resulted in a time series of 196 mosaiced images from 6 December 2013 to 12 March 2020. Errors and uncertainties associated with lake area and depth retrieval methods for each sensor are thoroughly discussed in Moussavi et al. (2016, 2020), Pope et al. (2016), Williamson et al. (2018) and Fricker et al. (2020).

Due to temporally varying satellite paths and/or cloud cover, only 11 out of the 196 mosaiced images covered the entirety of our AOI (Table S1). Therefore, to be able to compare areas and volumes of surface meltwater on dates with incomplete AOI coverage, we first created a mask of all pixels that were wet on at least one of the 196 dates analysed from 2013 to 2020 (Williamson et al. 2018), hereafter called a 'maximum wetted area mask'. Second, we created a 'maximum volume mask' by assigning all wet pixels in the maximum wetted area mask a depth equal to the maximum water depth observed out of all 196 images. Finally, for each image mosaic with ≥ 10% cloud-free coverage of our AOI (113 image dates), we normalized their total area and total volume of meltwater to our entire AOI using the following approaches. For each mosaiced image, we calculated the total observed meltwater area as a fraction of the total wetted area mask for the equivalent area. This fraction was then multiplied by the total area of the maximum wetted area mask over the whole AOI. To normalize the meltwater volume to the AOI, we did likewise, but instead used the maximum volume mask.

### 3.4 Local weather station data

We analyse the only available local AWS data in order to investigate the possible atmospheric driver(s) of the exceptional melt event over the northern GVIIS in 2019/2020. Near surface (2 m) temperature, relative humidity, wind direction, and wind speed data are available from the BAS Fossil Bluff AWS (Fig. 1b, yellow star, location: -71.329 S, -68.267 W, 66 m a.s.l.), at 12-hour intervals from 1979 to 1999, and at 5 or 10 minute intervals from 2000 to 2020. However, significant data gaps are present between 2000 and early 2007.

First, we compare the 2019/2020 daily mean air temperatures with the daily mean temperatures from 1979 to 2020 (using 12-hour data at noon and midnight local time); i.e. the complete time period for which we also have microwave radiometer data. Second, for 2007 to 2020, which is when AWS data are available at a higher frequency and data gaps are minimal (six months in the total record were missing values, but these were < 15% of the expected total for each of those months), we use the 10 minute data to calculate the length of time (in hours) when surface air temperatures are continuously ≥ 0°C during each melt season. Although the air temperature measured at a height of 2-m by the AWS will vary slightly from that at the ice surface, for the purposes of this study, we assume these temperatures to be equivalent (Kuipers Munneke et al., 2012). We also consider the occurrence of foehn winds, which are warm, dry winds often produced on the leeward side of mountains (Cape et al., 2015) and commonly occur on the AP (Luckman et al., 2014; Elvidge et al., 2016). Over GVIIS, the steep topography that generates foehn flow is provided by Alexander Island. We analyse foehn wind occurrence using a modified version of a metric previously used over the Larsen C Ice Shelf (Wiesenekker et al., 2018; Datta et al., 2019), whereby a 'foehn condition' is considered to initiate when air temperatures increase by ≥ 1°C, wind speed increases by ≥ 1.5 ms⁻¹ and relative humidity decreases by ≥ 5%,

all relative to the previous time step. We use a wind speed threshold of 1.5 ms$^{-1}$ instead of the higher threshold of 3.5 ms$^{-1}$ used by Datta et al. (2019) for the Cabinet Inlet AWS to account for lower foehn wind speeds over the northern GVIIS, which result from the lower mean elevation of the mountains on Alexander Island compared to those on the AP west of Cabinet Inlet (van Wessem et a., 2015). This foehn condition is assumed to remain until the conditions (with respect to the period preceding the foehn condition) are no longer met. Finally, we also examine differences in atmospheric regimes (temperature, wind direction and speed) within each wind direction class (northeasterly, northwesterly, southeasterly and southwesterly).

We note that it is beyond the scope of this study for us to identify specific meso-scale drivers of this exceptional melt event, especially as 2019/2020 is not a record melt season for the AP as a whole (see Section 4.1), and regional climate models frequently struggle to resolve localized surface melt in regions with highly-variable topography (Van Wessem et al., 2015), such as is the case for the GVIIS and its surrounding higher terrain.

## 4. Results

### 4.1 Microwave radiometer-derived melt observations over the Antarctic Peninsula

For the AP, cumulative melt days in the 2019/2020 austral melt season are highest in the southwestern areas of the AP (including the Wilkins and George VI ice shelves), in addition to the northern area of the Larsen C Ice Shelf (Fig. 2b). In comparison, cumulative melt days in 2019/2020 are relatively low over the southern areas of the Larsen C. The spatially-averaged cumulative melt days in the 2019/2020 melt season over the entire AP is 47 days (Fig. 2b), which is 53% higher (Fig. 2c) than the spatially-averaged climatology from 1979/1980 to 2019/2020 (31 days; Fig. 2a). However, of these 41 melt seasons, the 1992/1993 melt season has the highest spatially averaged cumulative melt days over the AP (62 days; Fig. S1). During that 1992/1993 season, although cumulative melt days over the northern GVIIS were only slightly higher than the 1979/1980 to 2019/2020 mean (Fig. S1d), cumulative melt days on the Larsen C Ice Shelf were particularly high, with a maximum of 117 cumulative melt days in the southern area of this ice shelf (Fig. S1c). This finding is contrary to the results of Bevan et al. (2020), who report that Larsen C experienced a 41-year record high melt year in 2019/2020. Bevan et al's (2020) results are based on microwave radiometer (SMMR/SSMI) data for melt seasons from 1979/1980 until 2016/2017, followed by microwave scatterometer (ASCAT) data from 2017/2018 to 2019/2020. In contrast, we use SSMR/SSMI data over the AP for the full 1979 to 2020 period to preserve consistency. As we explain in Section 3.2, ASCAT C-band radar is likely to be more sensitive to melt at depth than microwave radiometers, thus resulting in Bevan et al's (2020) higher calculated melt over Larsen C in the 2019/2020 season when combining data sources into one time series.

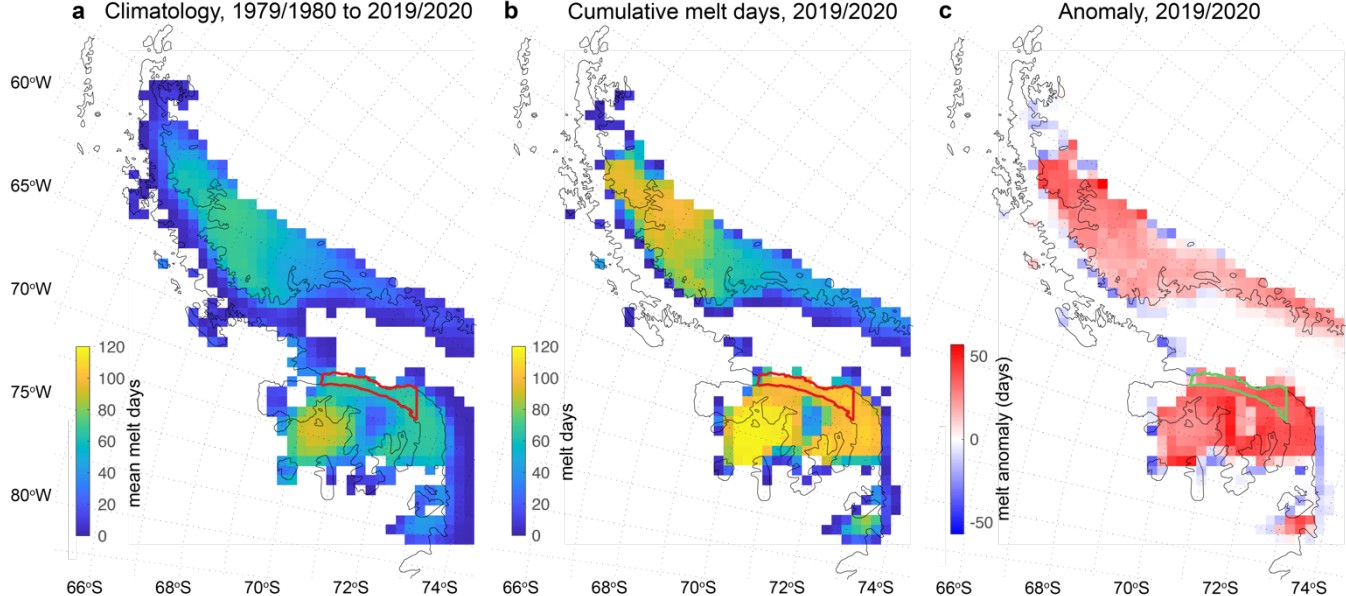

**Figure 2**. Microwave radiometer-derived maps of surface/near-surface melt days over the AP. (**a**) the climatology (i.e. mean cumulative melt days per season) from 1979/1980 to 2019/2020 (excluding 1987/1988 due to missing data); (**b**) cumulative melt days in 2019/2020; (**c**) the 2019/2020 melt season anomaly (i.e. (b) minus (a)). Melt days are counted within the period 1 November to 31 March (inclusive) each austral summer. The location of the study AOI is shown by a red outline in panels a) and b), and as a green outline in panel c). The black outline of the AP is from the MODIS Mosaic of Antarctica (Haran et al., 2014).

## 4.2 Microwave-derived melt observations over the northern GVIIS

Over the northern GVIIS, microwave radiometer-derived spatially-averaged cumulative melt days over the study AOI (12 grid cells, total area = 7556 km$^2$) in the 2019/2020 austral melt season is 101 days (Figs. 3b and d), which is higher than for any other melt season since the record began in 1979/1980, and is 53% higher (Fig. 3c) than the spatially-averaged climatology (66 melt days) from 1979/1980 to 2019/2020 (Figs. 3 and S2). However, as 41 days of microwave radiometer data for the 1987/1988 season are missing, we only conclude that 2019/2020 was the most significant melt season over 32 years (Fig. 3d). This result is supported by the analysis of scatterometer-derived melt data from ASCAT, which show that spatially-averaged cumulative melt days over the study AOI in the 2019/2020 austral melt season is 117 days; 70% higher than the spatially averaged climatology (69 melt days) from 2007/2008 to 2019/2020 (Fig. 4). The microwave radiometer-derived data suggests that 1989/1990 has the second highest spatially-averaged cumulative melt days (93) over the study AOI (Figs. 3d and S2).

Using the microwave radiometer data to consider the cumulative days of melting occurring over 100% of the AOI each season, 2019/2020 also sees the highest such number of days, 93 (Fig. 3d, dark blue bars), and 1989/1990 sees the second highest number of days (85). These values can be compared to a mean value of 53 cumulative melt days over 100% of the AOI from

1979/1980 to 2019/2020. Note that for each season, we do not specifically consider the mean areal extent of melting as this variable is found to be almost directly proportional ($r^2 = 0.9973$) to the spatially-averaged cumulative melt days (Fig. S3).

In terms of intra-annual patterns in percentage melt area over the northern GVIIS in 2019/2020, the microwave radiometer data shows that 100% of the AOI area experiences melting every day from 24 November 2019 to 22 February 2020 (Fig. 5c). After 22 February, the area of melting drops to 0% of the AOI over just three days, which is consistent with a drop in the mean daily air temperature (Fig. 5a). For a few weeks after 25 February, the area of melting fluctuates significantly, consistent with the air temperature fluctuating around 0°C. On 6 and 7 March 2020, 100% of the AOI is observed to melt again. From 16 March 2020, no additional melting is observed. Compared to the mean melt area over the two time periods shown in Fig. 5c (i.e. 1979/1980 to 2019/2020, and 2013/2014 to 2019/2020), the observed melt area in 2019/2020 covers 100% of the AOI for a significantly longer continuous period (91 days) relative to any other year in this record.

Since addressing uncertainties associated with microwave data products of binary melt/no melt information is challenging, this study uses two distinct microwave remote sensing techniques and algorithms to build further confidence in our conclusion. Moreover, our analysis of the sensitivity of the microwave radiometer (Fig. S4) and scatterometer (Fig. S5) melt detection algorithms to decreasing/increasing their threshold values shows that the 2019/2020 melt season remains exceptional, and a 32-year record.

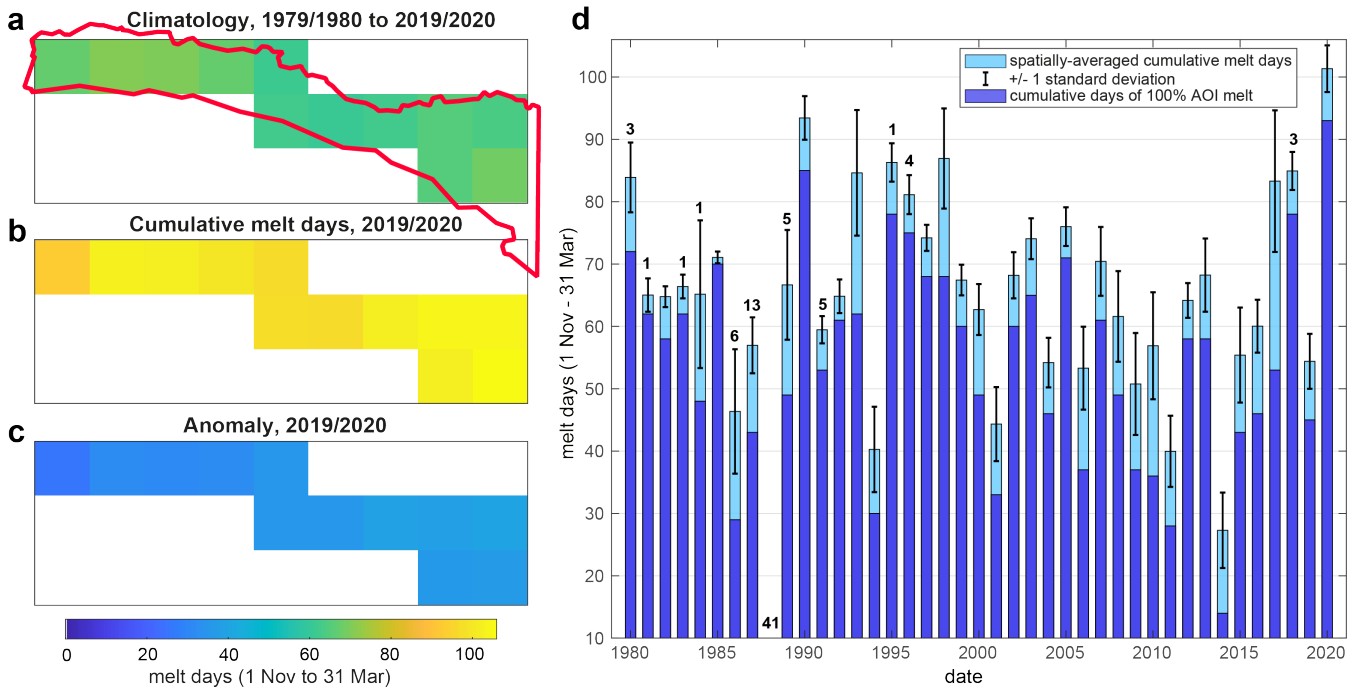

**Figure 3.** Microwave radiometer-derived cumulative melt days over the northern GVIIS AOI (see Fig. 1b for location, red outline) from 1 November to 31 March. (**a – c**) Maps of surface/near-surface melt days per 25 km grid cell. The relative location and shape of the study's AOI is shown by the red outline. White cells are out of the AOI. (**a**) Mean cumulative melt days for each 25 km grid cell for austral summers from 1979/1980 to 2019/2020, apart from 1987/1988 due to data unavailability. (**b**) Cumulative melt days per grid cell in the 2019/2020 austral summer. (**c**) Anomaly of the 2019/2020 melt season (i.e. (b) minus (a)). (**d**) Light blue bars represent spatially-averaged (i.e. over the 12 grid cells in the AOI) cumulative melt days, for each austral summer from 1979/1980 to 2019/2020 (apart from 1987/1988). The x-axis dates indicate the second year of each austral summer, e.g., 2020 corresponds to the 2019/2020 season. Black error bars show +/- one standard deviation from the spatially-averaged cumulative melt days. Dark blue bars show cumulative days when the melt extent is 100% of the AOI for each summer from 1979/1980 to 2019/2020. For melt seasons with missing data, the total number of missing data days is indicated by the black number above the corresponding bar.

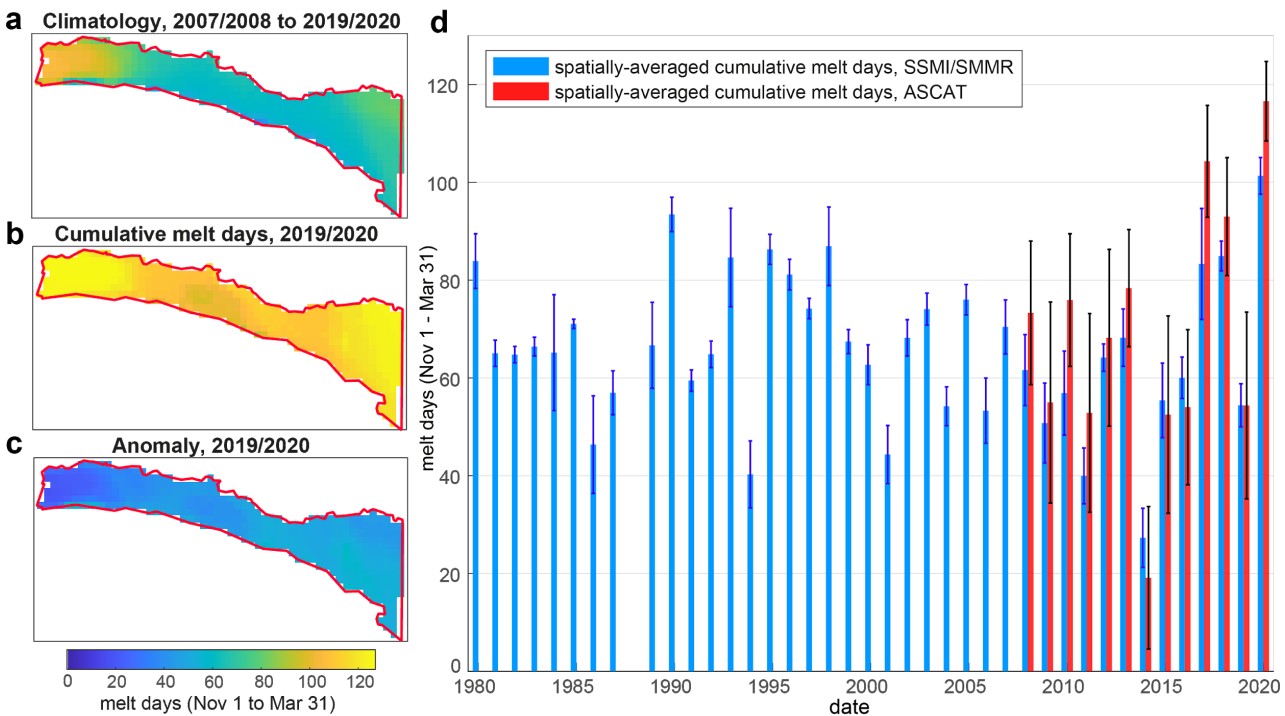

325

**Figure 4.** (**a – c**) Active microwave (i.e. ASCAT) derived cumulative melt days over the northern GVIIS AOI (see Fig. 1b for location, red outline), relative to the passive microwave time series (Fig. 3a - c). **a**) Mean cumulative melt days for austral summers from 2007/2008 to 2019/2020. **b**) Cumulative melt days in the 2019/2020 austral summer. **c**) Anomaly of the 2019/2020 melt season (i.e. (b) minus (a)). **d**) Red bars represent microwave scatterometer-derived spatially-averaged (i.e. over the AOI) cumulative melt days, for austral summers from 1979/1980 to 2019/2020 with red error bars showing +/- one standard deviation from the mean. Blue bars show microwave radiometer-derived spatially-averaged cumulative days from 1979/1980 to 2019/2020, with blue error bars showing +/- one standard deviation from the mean. The x-axis dates indicate the second year of each austral summer, e.g., 2020 = 2019/2020

### 4.3 Optical image-derived meltwater areas and volumes over the northern GVIIS

From 2013 to 2020, when we have Landsat 8 and/or Sentinel-2 optical imagery available, the day with the maximum observed area ($1.2 \times 10^9 \, m^2$) and volume ($6.2 \times 10^8 \, m^3$) of ponded surface meltwater on the northern GVIIS was 19 January 2020 (Figs. 1b, 5b, S6 and S7, Table S1), when 23% of the AOI is covered in ponded water. On this date, it is fortuitous that the whole of our AOI is visible in a mosaic of cloud-free Sentinel-2 image scenes (Fig. 1b; background image), and is also fully visible in a mosaic of Landsat 8 images acquired on 17 and 19 January (not shown). Calculated areas and depths of meltwater lakes on 19 January 2020 over the entire AOI are shown in Fig. S6. The mean depth of all water-covered pixels on this date is 0.52 m, and the maximum depth is 3.9 m. Unlike on other dates with much cloudier imagery, normalization of meltwater areas and volumes to the AOI on 19 January 2020 is not required (see Section 3.3 for method details, Fig. S7 for plots of both the observed and normalized meltwater areas and volumes for the 2013/2014 to 2019/2020 melt seasons, and Table S1 for details of all optical imagery analysed).

In all the seven melt seasons analysed with optical imagery, ponded surface meltwater volumes do not peak until January or February (Fig. 5b). However, in 2019/2020, meltwater volumes start to increase rapidly in late December/early January, which is earlier than in any other season, and corresponds with above average air temperatures in late December 2020 (Fig 5a, also see Section 4.4 for analysis of local weather conditions). In 2019/2020, volumes of meltwater ponding are highest in early January and then again in early February; corresponding with periods when mean daily air temperatures are $\geq 0^{\circ}C$ for extended periods (Fig. 5a). There is a notable decrease in surface meltwater ponding volume in mid to late January 2020 during a period of substantially colder air temperatures (i.e. $< 0^{\circ}C$, Fig. 5a) that likely resulted in widespread refreezing of surface meltwater (Fig. 5b).

The second largest melt season in terms of meltwater ponding is 2017/2018, with a peak in total meltwater area ($4.6 \times 10^8 \, m^2$) and volume ($2.5 \times 10^8 \, m^3$) on 29 January 2018 (Figs. 5b and S7). However, these two values are less than half of the respective values measured on 19 January 2020. Aside from 2019/2020 and 2017/2018 (i.e. the melt seasons with the greatest and second greatest volumes of surface ponding respectively), the other five melt seasons have relatively low volumes of ponded meltwater.

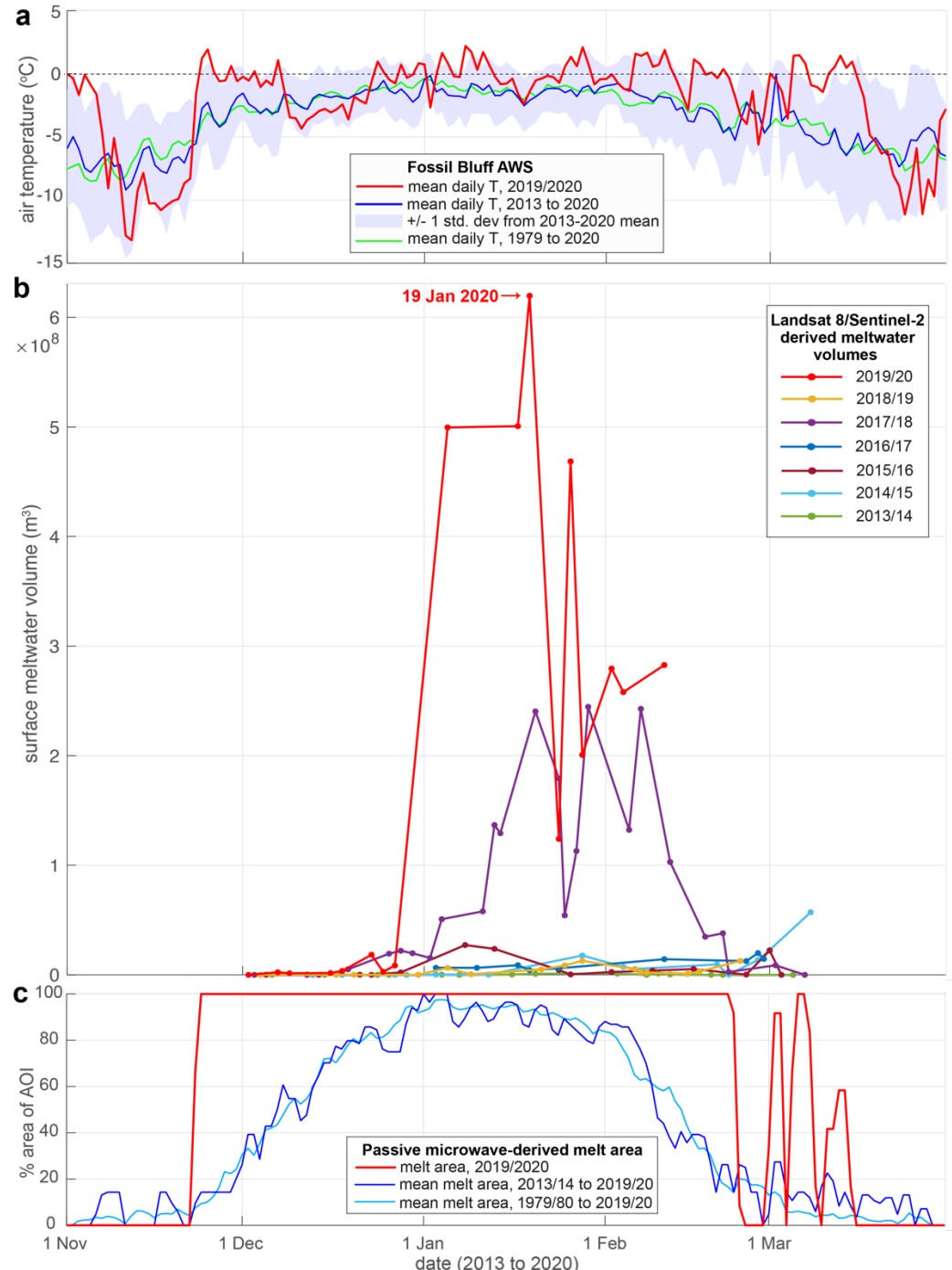

**Figure 5 (a)** Surface (2 m) air temperature data from the Fossil Bluff AWS (location in Fig. 1b, yellow star). The daily mean air temperature for the 2019/2020 melt season is shown by the red line, the daily mean temperature for the seven melt seasons from 2013/2014 to 2019/2020 are shown by the blue line, +/- one standard deviation from that blue line is shown by the areas

of blue shading, and the daily mean temperature from 1979 to 2020 (using 12-hour data) is shown by the green line. The horizontal black dashed line depicts 0°C. (**b**) Calculated volumes of surface meltwater ponding in the GVIIS AOI from 2013/14 to 2019/20, derived from Landsat 8 and Sentinel-2 optical imagery. Data from mosaiced images are only plotted if the image includes >10% of the study's AOI (Fig. 1, red outline) that is cloud free; data from mosaiced images on 113 images total are shown. On dates when imagery does not cover 100% of the AOI, observed meltwater volumes are normalized to the AOI (see Section 3.3 for further details, and Fig. S7 for a plot of all the observed meltwater volumes). (**c**) Microwave radiometer-derived near-surface melt extent over the GVIIS AOI (Fig. 1b, red outline) as a % of the total area (7556 $km^2$). Daily areas of melting for 2019/2020 are shown by the red line, the daily mean area of melting from 2013/2014 to 2019/2020 is shown by the dark blue line, and the daily mean area of melting from 1979/1980 to 2019/2020 (excluding 1987/1988) is shown by the light blue line.

## 4.4. Near-surface atmospheric conditions

Analysis of the mean daily surface air temperatures (derived from 12-hour values) from the Fossil Bluff AWS from 1979 to 2020 indicates that 2019/2020 is anomalously warm over five multi-day periods starting in late November (Figs 5a and S8). During these periods, mean daily air temperatures are ≥ 0°C for sustained time periods of up to a week. The total positive degree days for the 2019/2020 melt season (1 November to 31 March inclusive) is 40, compared to a mean of 19 ± 14 days (mean ± 1 standard deviation) from 1979/1980 to 2019/2020.

We also analyse the high-resolution (10 minute) AWS data from 2007 to 2020 to identify periods of sustained high temperatures, when it is possible that no refreezing at all occurred during the diurnal cycles, potentially enhancing the surface melt-albedo feedback effect. We find that the longest continuous period when air temperatures are ≥ 0°C in 2019/2020 is 90 hours in early February (Fig. 6, cyan line, and Fig. 7, label C). The longest five such time periods in 2019/2020 are labelled A to E in Fig. 7, and it is notable that two pairs of periods, A and B, and C and D, are only separated by a matter of hours. The mean length of these five longest periods when temperatures ≥ 0°C in 2019/2020 is 61 hours, which is longer than for any other season in the 2007 to 2020 high resolution AWS record (Fig. 6, black line). We also note that the temperature during these five periods is often more than one standard deviation greater than the multi-year daily mean (Fig. 5a). Considering all recorded temperature data in each melt season from 2007 to 2020, a higher percentage (33%) of 2019/2020 has air temperatures ≥ 0°C compared to any prior season (Fig. 6, red line).

We also examine the potential role of foehn winds on driving melt in 2019/2020. Foehn conditions (as described in Section 3.4) are only present for about 9 hours over the entire 2019/2020 season (Fig. 6, blue line), and occur in early and late summer (Fig. 7b, blue circles) when winds are typically stronger. We also note that the total time during each season when foehn conditions are calculated from AWS data has been relatively low since the 2007/2008 and 2008/2009 melt seasons, which each had a total of about 72 hours of foehn flow (Fig. 6). Therefore, foehn conditions do not appear to be dominant in driving melt in the 2019/2020 season.

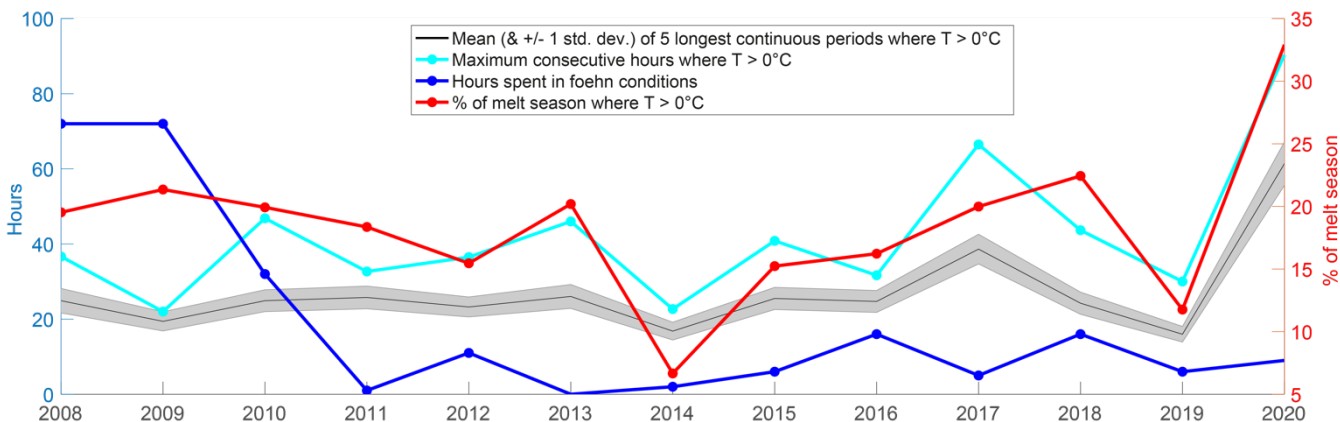

**Figure 6**. Analysis of sustained warm (≥ 0°C) air temperature (T) periods and foehn wind occurrence for the 2007/2008 to 2019/2020 melt seasons. The cyan line shows the maximum number of consecutive hours in each melt season when T ≥ 0°C. The black line shows the mean length (hours) of the five longest periods when T ≥ 0°C for each season, with the grey shading indicating +/- 1 standard deviation from that mean. The red line shows the proportion of each season (1 November to 31 March) when T is ≥ 0°C. The blue line shows the total number of hours spent in a foehn condition (see Section 3.4 for definition) each season. The x-axis dates indicate the second year of each austral summer, e.g., 2020 corresponds to the 2019/2020 season.

Finally, we analyse the potential role of warm air advection, resulting in sensible heat transport, on the high melt in 2019/2020. Considering wind direction alongside air temperature for melt seasons from 2007 to 2020, the climatology indicates that northwesterly winds dominate flow at all temperatures (Fig. 8a), but are more dominant when temperatures are ≥ 0°C (Fig. 8b), and are even more so when we limit analysis to just the five longest periods of sustained temperatures ≥ 0°C in each season (Fig. 8c). However, in the 2019/2020 season, northwesterly winds are less dominant (33% vs. 39%; Fig 8a), especially when only temperatures ≥ 0°C are considered (39% vs 47%; Fig. 8b), and further limited when only the five longest periods of sustained temperatures ≥ 0°C (Fig. 7; periods A-E) are considered (37% vs. 59%; Fig. 8c). Instead, the proportions of wind coming from the northeast are higher in 2019/2020 compared to the climatology (26% vs 24%; Fig 8a), particularly when temperatures are ≥ 0°C (32% vs 24%, Fig 8b).

The 2007 to 2020 climatology shows that, as expected, northwesterly winds typically include a higher proportion of warmer, faster winds, than other wind directions (Fig. S9a), whereas, northeasterly winds are typically lower-speed overall, and are generally colder (Fig. S9b). However, in the 2019/2020 melt season, we show that both northwesterly and northeasterly winds are warmer at lower wind speeds. Therefore, having eliminated foehn flow as a significant driver for surface melt in this season, we suggest that the increased advection of warm air from both the northwest and northeast contributed to the sustained warm air temperatures we observe.

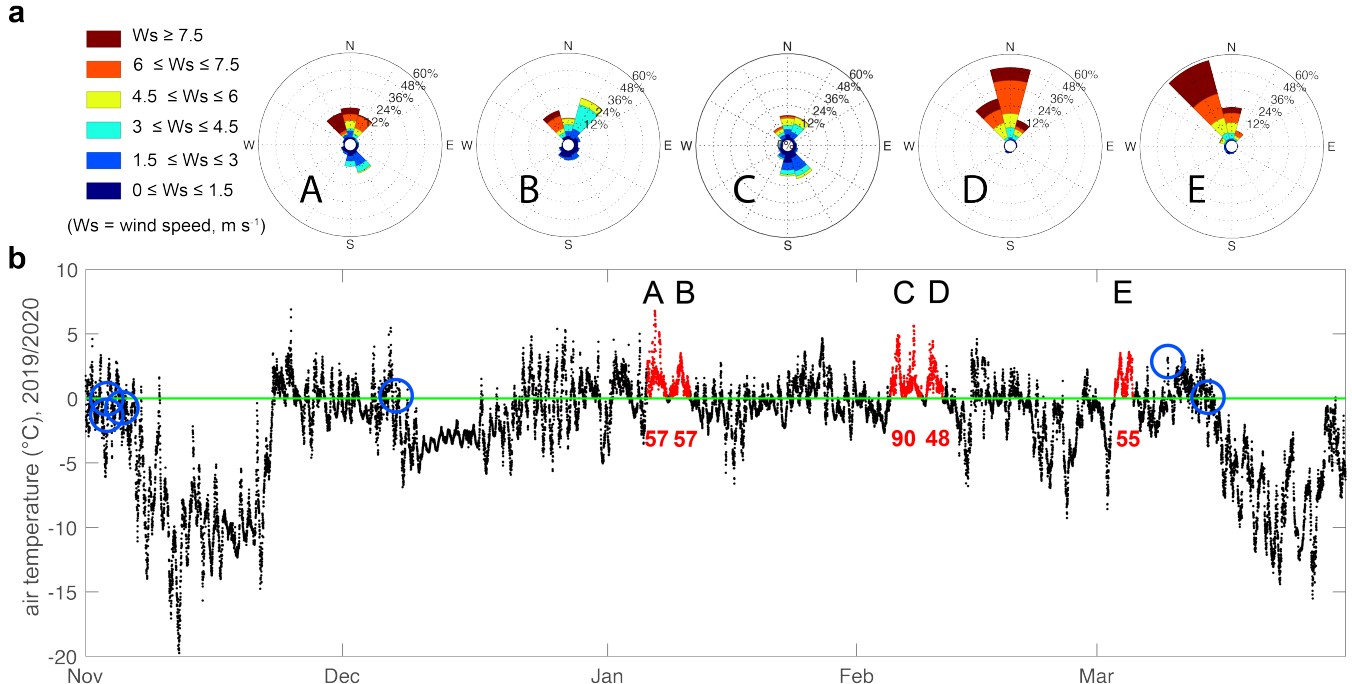

**Figure 7**. 2019/2020 wind and air temperature and data (10 minute) from the Fossil Bluff AWS. (**a**) Wind roses for corresponding periods of sustained air temperatures (A-E) indicated in b. (**b**) Air temperature record for 2019/2020 with the five longest periods of temperatures ≥ 0°C shown in red. The red numbers below these time periods indicate the total number of hours when the temperature is continuously ≥ 0°C. It is notable that only 9 hours separate periods A and B, and only 14 hours separate C and D. The six blue circles indicate periods when we calculate foehn conditions to be present (see Section 3.4 for methods).

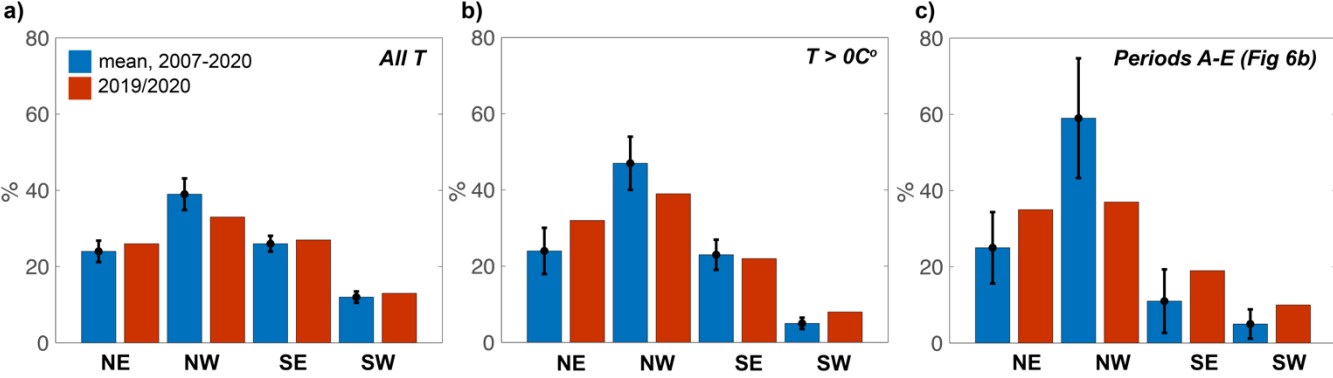

**Figure 8**. Percentage (%) of wind each season (1 November to 31 March) at Fossil Bluff AWS that is northeasterly (NE), northwesterly (NW), southeasterly (SE), southwesterly (SW), with the interannual (2007/2008 to 2019/2020) mean shown in blue and the 2019/2020 values shown in red. (**a**) Wind direction proportions using all recorded air temperatures. (**b**) Wind

direction proportions only when recorded temperatures are ≥ 0ºC. (**c**) Wind direction proportions only during the five longest periods of T ≥ 0ºC (A to E, Fig 7b) for all melt seasons (blue) and for 2019/2020 (red).


## 5. Discussion

### 5.1 Comparison of the optical image and microwave-derived melt data over the northern GVIIS, 2013 to 2020

Microwave melt data are binary (i.e. the algorithm indicates there is either melt or no melt), thus these data do not measure the
intensity of the melting, nor the volume of meltwater present. Additionally, while the optical data are used to detect the presence of surface meltwater, the microwave radiometer data can contain melt information through a snow depth of < 2 m, depending on the presence of surface lakes, and/or wetness of the subsurface snow/firn (see Section 3.1 for more detail). Therefore, we cannot directly compare the microwave-derived melt data with the optical image-derived ponding data. However, together, these data provide information on the time between melt onset and surface ponding over the northern GVIIS, and likewise the
disappearance of surface ponding and melt termination at the end of the season.

For the time period from 2013/2014 to 2019/2020, when we have three independent datasets, 2019/2020 was anomalous for the following reasons. Optical imagery indicates this season had the largest volumes of observed surface meltwater ponding (Fig. 5b), microwave radiometer- and scatterometer-derived data show that it also had the most spatially extensive melt (i.e. 100% of the AOI) for the greatest number of days (Fig. 3d), as well as the highest number of cumulative melt days (Figs 3d,
4d and S2).

In the 2019/2020 season, the microwave radiometer data first indicate the presence of surface/near-surface melt on 22 November, which extends to over 100% of the AOI by 24 November (Fig. 3c). However, surface meltwater ponding is not observed in the (non-continuous, both due to acquisition coverage and cloud coverage) optical imagery until mid-December (Fig. 5b). This offset in the timing of the observations is likely because although sustained positive air temperatures in late
November 2020 increased surface and near-surface melt rates, it takes time for surface ponds to develop in the early melt season, and this will only happen once suitable surface and firn/ice conditions are present. However, once surface ponds have developed, this offset in the timing between warm temperatures and ponding is much less apparent. For example, sustained warm temperatures in early January (Fig. 7b, periods A and B) and early February (Fig. 7b, periods C and D) coincide with periods when surface meltwater volumes derived from optical imagery are relatively high (Fig. 5b). Towards the end of the
melt season, although there are no cloud-free Landsat 8 or Sentinel-2 images available after mid-February 2020 (Fig. 5b), our visual analysis of Terra and Aqua MODIS optical imagery suggests that open water lakes remain until at least 25 February, with some lakes potentially remaining until mid to late March. Meanwhile, the microwave radiometer-derived melt drops to zero by 25 February, but then fluctuates until mid-March (Fig. 5c); perhaps indicative of a melt/refreeze process.

### 5.2 Near-surface/surface melting over the northern GVIIS, 1979 to 2020

From 1979/1980 to 2019/2020 (excluding the 1987/1988 season), the microwave radiometer data show that 2019/2020 was the largest melt season over the northern GVIIS in terms of the most spatially extensive melt (i.e. 100% of the AOI) for the greatest number days (Fig. 5c), and the greatest number of cumulative melt days (Figs. 3 and S2); results that are corroborated by our scatterometer-derived melt data from 2007 to 2020 (Fig. 4). As mentioned in Section 3.2, scatterometer-derived cumulative melt days (117 days) are likely higher than those derived from the radiometer data (101 days; Fig. 4d) because C-
band radiation has a larger penetration depth, thus likely detects melt at greater depths (Weber Hoen and Zebker, 2000).

The microwave radiometer data suggest a slightly negative trend in cumulative melt days and areal melt extent (Figs. 2, 3d and S2) from the mid 1990s until ~2015/2016, which is consistent with negative near-surface air temperature trends over the AP until 2016, likely relating to oscillations in the Southern Annular Mode (SAM) (Picard et al., 2007; Turner et al., 2016). This temperature trend is in contrast to the years prior to the mid/late 1990s, when trends over the AP from available research
station AWSs were generally positive since the 1950s (though this is not apparent in our microwave radiometer-derived melt data).

## 5.3 Local climatic controls of the 2019/2020 melt event

Our air temperature analysis using both daily means (from 1979; Fig. S8), and higher temporal resolution (10 minute) data (from 2007; Fig. 7b), shows anomalously long time periods when air temperatures were continuously ≥ 0°C in 2019/2020.
Using the 10-minute data, the longest such period was 90 hours in 2019/2020, suggesting that no refreezing occurred during that time (Fig. 7b). Overall, 2019/2020 also had the highest proportion of an entire season (33%) when temperatures were ≥ 0°C (Fig. 6). We suggest that the sustained periods of warm temperatures, which started unusually early in the melt season, both initiated and enhanced melting in 2019/2020. The presence of just a small quantity of surface meltwater early in the melt season is especially important as this will have a disproportionately effect on overall surface melt production due the non-
linear melt-albedo feedback process (Trusel et al., 2015).

Compared to the 2007 to 2020 AWS record, the 2019/2020 austral summer experienced a lower proportion of northwesterly wind (Fig. 8), though these winds are warmer at lower speeds (Fig. S9a). Instead, the proportion of northeasterly wind was higher in 2019/2020 compared to the 2007 to 2020 climatology (Fig. 8), and these winds were also warmer at lower speeds (Fig. S9b). We therefore suggest that sensible heat transported by warm, lower speed, northwesterly and northeasterly wind
helped to drive melting in 2019/2020. We also note that a record high Indian Ocean Dipole (IOD) in the early part of the 2019/2020 melt season is discussed in Bevan et al. (2020) as a potential large-scale driver for warm, northerly surface winds on the western AP. However, as the Fossil Bluff AWS does not measure radiation, we cannot exclude the possibility that the high melt in the 2019/2020 was not partially attributable to enhanced longwave radiation (potentially resulting from cloud cover) and/or increased shortwave radiation (potentially resulting from an absence of cloud cover).

Although warm foehn winds are known to initiate periods of sustained melt and/or produce firn densification due to near surface melt and refreezing (Luckman et al., 2015), our analysis suggests that the 2019/2020 melt season experienced limited

foehn conditions (see Section 3.4) in the early, and then late, melt season (Fig. 7b). This timing is predictable foehn flow behaviour, e.g., over the Larsen C, foehn winds are strongest in winter, when wind speeds are generally higher (Wiesenekker et al., 2018; Datta et al., 2019). Our observation of minimal foehn wind conditions over the northern GVIIS in 2019/2020 is consistent with our observation of an overall decrease in the frequency of northwesterly winds (Fig. 8), which are typically responsible for foehn flow. As we do not find 2019/2020 to be a record melt season for the AP as a whole (see Section 4.1), we chose to focus on identifying local climate drivers of this exceptional melt event based on the observational record, rather than trying to establish large-scale atmospheric drivers. Additionally, we note that even regional climate models often struggle to resolve localized surface melt in regions with highly-variant topography such as the northern GVIIS (Barrand et al., 2013; Van Wessem et al., 2015).

## 6. Conclusions

We have used microwave radiometer data from 1979 to 2020, and microwave scatterometer data from 2007 to 2020, to show that the 2019/2020 austral melt season on the northern GVIIS was exceptional in terms of both cumulative melt days and areal extent compared to the previous 31 melt seasons since 1988/1989, and possibly since the beginning of the record; 1979/1980. We also used multi-spectral satellite imagery from 2013 to 2020 to show that the observed surface meltwater ponding on the northern GVIIS in 2019/2020 was also exceptional in areal extent and also estimated volume since at least 2013/2014.

Our analysis, based on the local weather data from the Fossil Bluff AWS, shows that sustained periods of warm ($\geq 0^\circ$C) temperatures from early in the season (late November) likely contributed to the exceptional 2019/2020 melt event. These periods of sustained warm temperatures were likely driven by sensible heat transported by warm northwesterly and northeasterly low-speed winds. Consistent with our finding that the proportion of northwesterly wind decreased in 2019/2020 compared to the 2007 to 2020 period, we only calculate a total of ~9 hours of foehn conditions for this season, which occurred in early and late summer. It is therefore notable that although the high melt event over the northern GVIIS is 2019/2020 was caused by warmer than average air temperatures, such local weather conditions were not foehn-driven.

Using Landsat 8 and Sentinel-2 satellite imagery, we observed the maximum volume of meltwater ponding on the northern GVIIS (7850 km$^2$) on 19 January 2020, when ~23% of this area was covered in surface lakes with a mean depth of 0.5 m. In comparison, only 10% of the 3200 km$^2$ area of the Larsen B Ice Shelf that disintegrated in 2002 was covered in surface ponds with a mean depth of 0.8 m (Banwell et al., 2014). However, unlike the relatively unconstrained, and therefore, extensional ice flow of the Larsen B Ice Shelf (e.g. MacAyeal et al., 2003; Scambos et al., 2004), GVIIS has dominantly compressive flow, enabling the shelf to remain relatively stable despite large volumes of surface water (Lai et al., 2020). Despite this, our results show that some of the areas of dense surface ponding near the eastern margin of the northern GVIIS coincide with areas classified as vulnerable to hydrofracture by Lai et al. (2020, their Fig. 4), particularly if pre-existing surface crevasses are present. Though individual years of exceptional high surface melt do work to decrease ice-shelf stability, further research is

required to better constrain the potential timing and style of a GVIIS collapse event due to the competing controlling factors of surface melt, basal melt, and stress regime.

530

*Code and data Availability.* The code used to calculate areas and volumes of surface meltwater is available at https://github.com/mmoussavi/Lake_Detection_Satellite_Imagery/ and described in detail in Moussavi et al. (2020). A comprehensive dataset of Antarctic lakes from Landsat 8 imagery (Moussavi, 2020) is also available at https://doi.org/10.15784/601401. The passive microwave melt product is available at http://pp.ige-grenoble.fr/pageperso/picardgh/melting/, last accessed on October, 1 2020. The ASCAT enhanced resolution product is available at https://www.scp.byu.edu/data/Ascat/SIR/msfa/Ant.html, last accessed on October 1, 2020. Temperature data are available from the BAS Fossil Bluff AWS at 10 minute intervals from 2006 to 2020 here: https://legacy.bas.ac.uk/cgi-bin/metdb-form-2.pl?tabletouse=U_MET.FOSSIL_BLUFF_ARGOS&complex=1&idmask=.....&acct=u_met&pass=weather), and at intervals ranging from 12 hour to 1 hour from 1979 to 2006 here: https://legacy.bas.ac.uk/cgi-bin/metdb-form-2.pl?tabletouse=U_MET.FOSSIL_BLUFF_SYNOP&complex=1&idmask=.....&acct=u_met&pass=weather)

*Author contributions.* The authors are ordered by their relative levels of contribution. AFB conceived the study, analysed the optical image-derived and microwave-derived melt data, and drafted the manuscript. RTD performed the local climate analysis. MM derived surface meltwater areas and volumes from the optical satellite imagery. RD further processed the optical imagery-derived results. LB and GP processed the microwave radiometer and scatterometer data to produce the melt data. CAS identified key Landsat 8, Sentinel-2 and MODIS image dates for analysis. All authors discussed the results and were involved in editing of the manuscript.

*Competing interests.* The authors declare that they have no conflict of interest.

*Acknowledgments.* AFB received support from the U.S. National Science Foundation (NSF) under award #1841607 to the University of Colorado Boulder, and from a CIRES Postdoctoral Visiting Fellowship. RTD was funded by the NASA ICESat-2 Project Science office. RLD was funded by a Natural Environment Research Council (NERC) Doctoral Training Partnership Studentship (CASE with the British Antarctic Survey, #NE/L002507/1). MM was funded by NSF GEO award #1643715 to the University of Colorado Boulder. LB and CAS were funded by the NASA Cryospheric Science Program. GP was funded by the European Space Agency (ESA) project 4D Antarctica (ESRIN:4000128611/19/I-DT). LAS received support from the U.S. NSF under award #1841739 to Columbia University. The authors thank Steve Colwell at the British Antarctic Survey (BAS) for help with the BAS Fossil Bluff AWS data acquisition. Doug MacAyeal, Ian Willis, Ted Scambos and Julie Miller are all thanked for useful discussions, and Julie Miller is also thanked for producing the MODIS mosaic in the background of Fig 1a.

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
