# Peer review of "32-year record-high surface melt in 2019/2020 on the northern George VI Ice Shelf, Antarctic Peninsula"

_The Cryosphere, 2020_

## Referee Comment (RC1) · Anonymous Referee #1 · 20 Nov 2020

**32-year record-high surface melt in 2019/2020 on north George VI Ice Shelf, Antarctic Peninsula**

Banwell et al.

Anonymous referee

**General comments:**

This manuscript presents a quantitative analysis of surface meltwater on the George VI Ice Shelf, Antarctic Peninsula, focusing on the most recent melt season (2019/2020) and setting this in the longer-term record of melt. The authors use data from a number of sources, including microwave radiometer and scatterometer, automatic weather station measurements and a previously published algorithm to classify surface meltwater ponding from Landsat 8 and Sentinel-2 imagery.

Meltwater has been linked to the instability and collapse of Antarctic Peninsula ice shelves. Although the focus and methods of this study are not novel, it quantifies the recent anomalous melt event on this ice shelf, which is important given that such record-high melt events are set to become more frequent in future. Therefore, it is my view that the findings from this manuscript are of broad interest to the cryospheric community.

In general, I would like to complement the authors on their well-written and clearly-structured manuscript, and the study rationale and methods are well-justified. The results build upon previous work that has reported surface meltwater lakes on this ice shelf by providing a time series of ponded surface meltwater together and analysing this alongside microwave observations of melt together with local climatic controls.

The authors use optical Landsat 8 and Sentinel-2 imagery to derive ponded meltwater volumes. However, I wonder why historical satellite imagery pre-2013 was not used to supplement this record and set 2019/2020 within the longer-term context of surface ponding? I suspect this may be related to difficulties applying the lake depth radiative transfer model to historical imagery, but it would be worth justifying.

There could be more of a discussion on the fate of surface meltwater during and at the end of the melt season. For example, is the decrease in meltwater volume in mid-late January associated with refreezing, or is there any evidence of englacial lake drainage? Similarly, do the authors observe any rapid drainage events, and if so at what point in the melt season?

There is also a lack of discussion of uncertainties, especially melt detection uncertainty using the microwave brightness temperature product and the ASCAT product. In addition, the reader should be made aware of the limitations in using the depth retrieval algorithm applied to optical satellite imagery (see Sneed and Hamilton, 2011 and Pope et al., 2016).

I think it could also be highlighted more clearly in the manuscript that this record surface melt in 2019/2020 was unrelated to foehn-driven melting (see specific comments).

Once the authors address these issues and my specific comments below, I can therefore recommend that this manuscript is suitable for publication in The Cryosphere.

**Specific comments:**

Line 25: consider quantifying 'low-speed' winds here, i.e. $\leq 7.5$ ms$^{-1}$

Line 63: Consider adding either an additional sentence here or an additional panel in Figure 1 showing which other ice shelves experienced increased meltwater ponding in 2019/2020 to provide further context. Although it is mentioned in the Figure caption that Larsen C, Wilkins and Bach also experienced ponding, this is not immediately clear from Panel A.

Line 66: I suggest either enlarging the latitude-longitude labels on Panel A, or adding them to Panel B. In addition, I think it would be helpful to add an arrow labelling ice flow direction.

Line 87: Quantify surface summer melt rates (e.g. up to ~400 mm w.e. yr$^{-1}$, Trusel et al., 2013).

Line 90: Reynolds (1981) discusses observations of moulins on George VI, so consider modifying this sentence.

Line 91: Quantify 'high' basal melt rates and thinning rates.

Line 93: Consider adding one line in this paragraph quantifying ice flow speeds on northern GVIIS, since ponding preferentially occurs on slower-moving ice.

Line 95: I suggest also citing Lucchitta and Rosanova (1998) here as well.

Line 99: I suggest making it clearer in this paragraph that these three types of surface lakes form every austral summer to make it clearer you are not just referring to 2019/2020. In addition, consider adding sub-panels to Figure 1 to show examples of these three types, or a separate Supplementary Figure.

Line 100: I suggest you also cite Hambrey and Dowdeswell (1994) here, and perhaps add to the end of this sentence that ice flowlines are surface manifestation of longitudinal foliation.

Line 104: Please consider also citing Langley et al. (2016) and Arthur et al. (2020b) here, which record observations of down-ice lake advection on Langhovde Glacier and Shackleton Ice Shelf, East Antarctica.

Line 111: I suggest continuing the final sentence with: 'as described above, enabling it to support a large surface area of surface meltwater (Alley et al., 2018).

Line 117: Be consistent with the use of northern/north GVIIS; I think northern is used most frequently throughout.

Line 125: What is the uncertainty associated with this microwave brightness temperature product and the ASCAT product in Section 3.2? There could be more discussion of this.

Line 129: I suggest adding one sentence after this along the lines of: the sensor measures the emitted energy from the surface and sub-surface, proportional to the brightness temperature (which increases with the presence of liquid water and which increases absorption and emissivity).

Line 142: Briefly state why grid cells > 1700 m a.s.l. were masked out, presumably to only show data corresponding to ice shelf areas.

Line 148: Consider adding a citation here?

Line 161: I suggest adding a sentence explaining where this threshold comes from i.e. was based on empirical comparisons with QuikSCAT-derived melt (Ashcraft and Long, 2006).

Line 166: Add 'scatterometer-derived' before 'cumulative melt days'.

Line 177: Perhaps briefly outline here the threshold-based algorithms other than NDWI that this method uses, i.e. NDSI and others, or else list which bands are used.

Line 182: Consider also citing Bell et al. (2017) and Arthur et al. (2020b) here.

Line 184: This depth-reflectance algorithm makes a number of assumptions which I think are worth briefly outlining in an additional sentence, including the assumption of homogenous lake bottom albedos, minimal wind-driven light scattering, etc (Sneed and Hamilton, 2007).

Line 185: There is no discussion currently in this section of false positives. How did you deal with these (if there were any), and were they manually removed?

Line 218: I think 'warm' here more accurately describes foehn winds than 'hot'. Also suggest adding at the end of this sentence 'and commonly occur on the AP', and citing Luckman et al. (2014).

Line 219: I suggest adding an indication here of how steep the topography is, e.g. maximum slope.

Line 339: This is an interesting finding. I wonder whether it would be worth adding an additional sentence (either here or at the end of this section) explicitly summarising these observations demonstrate you can still get record melt when conditions are generally warmer, with no involvement of foehn wind events.

Line 393: Are there any measurements of firn air content/thickness on George VI?

Line 410: I don't think you can necessarily suggest that surface melt volumes were highest in 2019/2020 out of the prior 31 seasons without having explicitly derived volume estimates from imagery pre-2013.

Line 419: Change 'zero re-freeze' to 'no refreezing occurred'.

Line 434: Explain 'shoulder seasons' – do you mean colder seasons (autumn/winter?).

Line 446: Change 'back to 2013' to 'from 2013 to 2020' and suggest adding to the last part of the sentence: 'was also exceptional in areal extent and volume [..]'.

Line 454: Add AOI area here in brackets to remind readers of its size in comparison with Larsen B in the next sentence. Also, what was the maximum lake depth?

Line 458: I suggest also citing Alley et al. (2018) here.

Line 460: Consider showing in a Supplementary Figure the lakes mapped in this study overlaid onto the areas classed as vulnerable to hydrofracture by Lai et al. (2020).

Figure S5: I notice there are two particularly deep lakes – why, out of interest, do you suggest this is?

**Technical corrections:**

Line 31: minor point, but check here and throughout that citations are listed chronologically.

Line 49: Add comma after 'However'.

Line 100: 'most extensive' rather than 'the largest'?

Line 106: Change en-echelong to '*en-echélon'* and consider adding a brief description for those unfamiliar with this term, e.g. closely spaced, sub-parallel.

Line 120: Change '(from 2013)' to 'from 2013-2020' and the same with '(from 2017)'.

Line 125: Insert 'passive' before 'microwave radiometer'.

Line 132: Hyphenate 25 km.

Line 157: Hyphenate 4.45 km and consider briefly outlining the SIR algorithm.

Line 159: Hyphenate 'SMMR-based'.

Line 173: Rephrase to 'to selected multispectral imagery (see paragraph below)'.

Line 197: Are italics needed in this paragraph?

Line 216: Remove 'periods'.

Line 220: citation should be Wiesenneker et al., 2018 (full reference is correct).

Line 291: remove commas after 2020 and AOI.

Line 327: Change 'refreeze' to 'refreezing'

Line 390: Re-word sentence to 'meltwater ponding is not observed in the optical imagery until mid-December'.

Line 495: Please add volume, issue and page numbers: 44(6), 837-869.

**References:**

Alley KE, Scambos TA, Anderson RS, et al. (2018) Quantifying vulnerability of Antarctic ice shelves to hydrofracture using microwave scattering properties. *Remote Sensing of Environment* 210: 297–306.

Arthur, J. F., Stokes, C. R., Jamieson, S. S. R., Carr, J. R., and Leeson, A. A.: Distribution and seasonal evolution of supraglacial lakes on Shackleton Ice Shelf, East Antarctica, *The Cryosphere Discuss.,* https://doi.org/10.5194/tc-2020-101, 2020b.

Hambrey MJ and Dowdeswell JA (1994) Flow regime of the Lambert Glacier-Amery Ice Shelf system, Antarctica: Structural evidence from Landsat imagery. *Annals of Glaciology* 20: 401–406

Langley ES, Leeson AA, Stokes CR, et al. (2016) Seasonal evolution of supraglacial lakes on an East Antarctic outlet glacier. *Geophysical Research Letters* 43(16): 8563–8571.

Lucchitta BK and Rosanova CE (1998) Retreat of northern margins of George VI and Wilkins Ice Shelves, Ant- arctic Peninsula. *Annals of Glaciology* 27: 41–46.

Luckman A, Elvidge A, Jansen D, et al. (2014) Surface melt and ponding on Larsen C Ice Shelf and the impact of fohn winds. Antarctic Science 26(6): 625–635.

Reynolds RW (1981) Lakes on George VI Ice Shelf. *Polar Record* 20(128): 425–432.

Sneed WA and Hamilton GS (2007) Evolution of melt pond volume on the surface of the Greenland Ice Sheet. *Geophysical Research Letters* 34(3): 4–7.

---

## Referee Comment (RC2) · Anonymous Referee #2 · 25 Nov 2020

This is an engaging study investigating the 2019/2020 austral melt season on the north George VI Ice Shelf, Antarctic Peninsula. The authors combine different datasets such as satellite microwave radiometer/scatterometer data, optical satellite imagery and local meteorological station data. The authors highlighted exceptional melt and surface ponding conditions in 2019/2020, and they concluded that warm temperatures were likely triggered by warm northwesterly and northeasterly low-speed winds, rather than Foehn winds.

The paper is well written, and the results are supported by the data and methods. The study has the potential to become a fully relevant article to concept of The Cryosphere

and my criticisms are with regards presentation rather than science.

Specific comments:

Lines 31-32: Could you please specify the rate of mass loss?

Lines 53-54: Is there any projected estimation of contribution of the AP ice shelf melting to the global sea level rise? If yes, that information would be useful.

Lines 56-58: I would add a link of official WMO status on this evaluation:

https://public.wmo.int/en/media/news/new-record-antarctic-continent-reported

In addition, to my knowledge, WMO is evaluation the value of 18.4°C., not the 20.75°C. There has been no official effort/information to evaluate the value of 20.75°C. Furthermore, this value was not recorded at the official Marambio station. I recommend the authors to make sure that whether this temperature is being evaluated or not. If not, I recommend them to remove the statement in lines 58-60.

Lines 60-63: Could you please quantify melt amount in this part too? Because, you are comparing GVIIS with the other AP ice shelves. Some numerical values would help the reader to see the differences.

Lines 87-94: What makes the northern GVIIS more vulnerable to high surface summer melt rates? I would expect to consider rates of basal melting, nonetheless, you state that rates of basal melting are greatest at the southern end of the GVIIS. Therefore, a clarification might be needed to have a better idea on different physical processes at the northern and southern ends.

Lines 123, 208-212: Is there any elevation difference between the AWS station and ice shelf?

Line 213: What does it mean exactly "using 12-hour data" for the 1999-2020 period? I understand that you use only one time step as a daily mean temperature for the 1979-1999 period. If yes, what is the exact hour of the observation (morning, late afternoon)?

For the second period, did you pick up the same hour with the 1999-2020 period or did you take a temporal mean of 12-hour data?

Line 223: Could you please specify why there exist lower mean speeds over north GVIIS?

Lines 238-239: In your analysis, I understand that Larsen C shows a record high melt year in 1992-1993, not in 2019-2020 on the contrary to the findings of Bevan et al. Is that related just with the different datasets? Can you please specify potential uncertainties of each product?

Lines 326-334: Why not to compare the longest warm periods with the volume changes? For instance, are the longest periods coincide with the largest volume change?

Lines 335-339: I think the authors should also discuss the potential role of the warm air advection given the low foehn conditions. I suspect that regional sensible heat transport seems to be one of the main contributors of the 2019/2020 melt season, particularly for the first two weeks of February.

Lines 349-359: Can you specify where the calculated 9 hour of foehn is located in this time series (Fig. 6)?

Lines 431-438: I am a bit disconnected here. So, I understand that rather than local conditions (i.e., Foehn winds) one should expect to see regional- and/or large-scale warm advection. In this case, I would expect to see similar warming rates over the GVIIS and Larsen C. However, in Fig. 2c there are notable warming differences between these two regions. Do the authors have any idea for these differences?

Figures:

Fig. 1 and Fig. 2: Could you please use a larger font size for the lat/lon coordinates?

Overall suggestion: As you use different datasets with different spatial and temporal

resolutions, I would add a table for the information given in data and methods (e.g., name of the product, resolution, temporal coverage etc.). This makes it easier for reader to follow the result.

---

## Referee Comment (RC3) · Anonymous Referee #3 · 26 Nov 2020

This is a well-written and evidenced paper that uses several independent datasets to explore the precise characteristics melting over the George VI Ice shelf. The authors demonstrate that melt extent and duration over GVIIS was higher in 2019/20 than in any other melt year analysed, and suggest that localised meteorological factors were responsible. Namely, persistent north-westerly and north-easterly low-speed winds, which allowed temperatures to rise above freezing and for melt to occur.

The manuscript is presented in a logical structure, with interesting and informative figures. The methods used are justified and largely given in sufficient detail to be reproducible. The results are supported by the data presented and the conclusions

drawn are consistent with the evidence given. I recommend the publication of this manuscript in TC subject to minor revisions.

Specific comments:

[L49-52] You make reference to the collapse of Larsen A (and Larsen B in a previous sentence) and then in the next sentence to the partial collapse of Larsen A and B – could you revise this to be clearer/more consistent with the above?

[L57-61] I think the 20.75 record has been thrown out by the WMO so you can probably remove this part of the sentence. The justification can be made using the Esperanza record alone.

[L62] You could link this back to the Bevan et al. paper again

[Section 3.4] Can you comment on the data processing for the AWS – e.g. is it an instantaneous value every 12 hours, or an average?

[L172] extra "to" after parenthesis

[para starting 187] Is there precedent for this type of methodology?

[para starting L320] It may be worth commenting on the fact that surface temperatures can differ from 2 m air temperatures and so 0 degrees at 2 m may not actually mean surface temperatures are at the melting point / vice versa. How might this influence the length of periods of >0 temperatures and consequently refreezing?

[para starting L335] During how much of the time series in Fig. 4a was the temperature above 1 standard deviation above the mean? This might be another interesting way to think about this anomalously warm period.

[para starting 335] Was any sensitivity testing conducted regarding the foehn detection method? Different thresholds and methods can yield very different results, e.g. isentrope-based method of King et al. (2017, doi: 10.1002/2017JD02680) vs surface method similar to the one used here. Varying the thresholds used can also make a

difference (e.g. Turton et al., 2018, doi.org/10.1002/qj.3284).

---

## Author Comment (AC1) · 8 Jan 2021

**Authors' response to anonymous Reviewer #1**

This manuscript presents a quantitative analysis of surface meltwater on the George VI Ice Shelf, Antarctic Peninsula, focusing on the most recent melt season (2019/2020) and setting this in the longer-term record of melt. The authors use data from a number of sources, including microwave radiometer and scatterometer, automatic weather station measurements and a previously published algorithm to classify surface meltwater ponding from Landsat 8 and Sentinel-2 imagery.

Meltwater has been linked to the instability and collapse of Antarctic Peninsula ice shelves. Although the focus and methods of this study are not novel, it quantifies the recent anomalous melt event on this ice shelf, which is important given that such record-high melt events are set to become more frequent in future. Therefore, it is my view that the findings from this manuscript are of broad interest to the cryospheric community.

In general, I would like to complement the authors on their well-written and clearly-structured manuscript, and the study rationale and methods are well-justified. The results build upon previous work that has reported surface meltwater lakes on this ice shelf by providing a time series of ponded surface meltwater together and analysing this alongside microwave observations of melt together with local climatic controls.

We thank this reviewer for their complementary remarks about our paper and we are pleased to hear that they think it will be of broad interest to the cryospheric community.

The authors use optical Landsat 8 and Sentinel-2 imagery to derive ponded meltwater volumes. However, I wonder why historical satellite imagery pre-2013 was not used to supplement this record and set 2019/2020 within the longer-term context of surface ponding? I suspect this may be related to difficulties applying the lake depth radiative transfer model to historical imagery, but it would be worth justifying.

Analysis of pre-2013 imagery would require tuning Moussavi et al's threshold-based algorithm for the Landsat 7 sensor (and for prior Landsat sensors). This could be done, however, with much of Landsat 7 data missing (due to SLC off gaps), lake volumes derived from this sensor would not be easily comparable to L8 and S2. In other words, the inclusion of Landsat 7 data will not necessarily extend the time series record, simply because its data is not comparable to Landsat 8 and Sentinel-2. We will briefly state these reasons in our revised paper.

There could be more of a discussion on the fate of surface meltwater during and at the end of the melt season. For example, is the decrease in meltwater volume in mid-late January associated with refreezing, or is there any evidence of englacial lake drainage? Similarly, do the authors observe any rapid drainage events, and if so at what point in the melt season?

This is a helpful suggestion and we will include some additional statistics in the main text (inc. mean depth and maximum depth of water-covered pixels). Additionally, we will add some sentences that describe links between the total volume of surface ponding and the air temperatures, particularly for the 2019/2020 season. However, to intercompare volumes of meltwater between different austral melt seasons, the optical image analysis part of this study focusses on temporal variations in the total area and volume of surface meltwater (and particularly the maximum values) each season. Therefore, we did not produce timeseries of data for individual lakes, and producing those data would involve re-running the algorithm and performing a substantial new amount of analysis that is beyond the scope of the current study. However, from visual analysis of the key images, we have not noticed any evidence

of rapid lake drainage events, and due to the compressive flow regime (see current lines 84-86), we also would not expect such events to be common.

There is also a lack of discussion of uncertainties, especially melt detection uncertainty using the microwave brightness temperature product and the ASCAT product.

Addressing uncertainties associated with microwave data products of binary melt/no melt information is challenging. In-situ measurements (e.g., of snow temperature, liquid water content) for true validation purposes are rare. Assessment against air temperature measurements have been done in the past. In our study, two distinct microwave remote sensing techniques and algorithms were used to bring confidence in our conclusion; we used both microwave radiometer (SMMR/SSMI) data and microwave scatterometer (ASCAT) data. Although the figure showing the ASCAT-derived results is currently in the Supplementary Information (current Fig. S3), we propose to move this figure into the main paper.

Additionally, to further address the reviewer's comment and to bring new content to our study, we have analyzed the sensitivity of the melt detection algorithms to their threshold values; the most uncertain part of the algorithms. Please see below for results of this analysis for both melt products, which confirm that our conclusion of the exceptional 2019/20 melt year is robust.

SMMR/SSMI uncertainty analysis

The figure below shows the brightness temperature temporal evolution for one pixel in North GVIIS (lat, lon = -72.287, -67.579) over the course of the 2019/2020 melt season. Three algorithm thresholds, defined by the mean brightness temperature plus the standard deviation multiplied by a coefficient, are also indicated. The coefficient of 3 is nominal (Torinesi et al. 2001) and was used our study, as in other past studies. For this single pixel, the resulting cumulative melt days for each threshold are: coef: 2.5 -> 107, coef: 3 -> 103, coef: 3.5 -> 99.

[Figure]

Using each of these three coefficients (2.5, 3, 3.5), we have also calculated the spatially-averaged cumulative melt days over the northern GVIIS for all melt seasons from 1979/1980 to 2019/2020 (apart from 1987/1988 due to missing data), the results for which are shown in the bar plot below. We estimate that changing the coefficient from 2.5 to 3.5 produces a change in the cumulative melt days in the order of 10%. Importantly, as shown by the plot below, the 2019/2020 melt season is exceptional regardless of the choice of coefficient used in the SMMR/SSMI analysis.

[Figure]

**ASCAT uncertainty analysis**

The bar plot below shows the ASCAT-derived spatially-averaged cumulative melt days over northern GVIIS for melt seasons from 2007/2008 to 2019/2020 using three thresholds; 2 dB, 3 dB (the value we use in our study), and 4 dB. As this plot shows, varying the threshold does not alter the conclusion that the 2019/2020 melt season experienced the highest cumulative melt days.

[Figure]

The revised manuscript will include the following sentences in Results section 4.2: "*Since addressing uncertainties associated with microwave data products of binary melt/no melt information is challenging, this study uses two distinct microwave remote sensing techniques and algorithms to build further confidence in our conclusion. Moreover, our analysis of the sensitivity of the melt detection algorithms to decreasing/increasing their threshold values confirm that the 2019/2020 melt season was a 32-year record.*" We will also include the two bar plots above in the Supplementary Information for our revised paper.

In addition, the reader should be made aware of the limitations in using the depth retrieval algorithm applied to optical satellite imagery (see Sneed and Hamilton, 2011 and Pope et al., 2016).

Regarding limitations associated with using the depth retrieval algorithm applied to optical images, in our revised manuscript, we will also add the following sentence about the assumptions the depth algorithm makes: "*This approach makes a number of assumptions, including that the lake bottom has a homogenous albedo, there is little to no particulate matter in the water column to alter its optical properties, and that there is minimal wind-induced surface roughness (Sneed and Hamilton, 2007).*"

I think it could also be highlighted more clearly in the manuscript that this record surface melt in 2019/2020 was unrelated to foehn-driven melting (see specific comments).

We agree with this comment and will add extra sentences to clarify this in the Abstract, Results and Conclusion sections.

Once the authors address these issues and my specific comments below, I can therefore recommend that this manuscript is suitable for publication in The Cryosphere.

Specific comments:

Line 25: consider quantifying 'low-speed' winds here, i.e. ≤ 7.5 ms$^{-1}$

We will do as suggested.

Line 63: Consider adding either an additional sentence here or an additional panel in Figure 1 showing which other ice shelves experienced increased meltwater ponding in 2019/2020 to provide further context. Although it is mentioned in the Figure caption that Larsen C, Wilkins and Bach also experienced ponding, this is not immediately clear from Panel A.

We thank the reviewer for these suggestions. We would prefer not to add an additional panel to figure 1 as that would mean that panel (b) would need to become smaller, meaning lakes on GVIIS would be less visible. Instead we plan to add the following additional sentence about ponding on the Wilkins, Bach and Larsen C to the main paper: "*However, as Fig. 1a shows, in 2019/2020, surface meltwater ponding was also prevalent on the northwestern Larsen C, the northern Wilkins (also visible in the bottom left corner of Fig 1b), and the northern and northwestern Bach ice shelves.*"

Line 66: I suggest either enlarging the latitude-longitude labels on Panel A, or adding them to Panel B. In addition, I think it would be helpful to add an arrow labelling ice flow direction.

We will enlarge the latitude-longitude labels on panel a) and also add an arrow to indicate ice flow on panel b).

Line 87: Quantify surface summer melt rates (e.g. up to ~400 mm w.e. yr, Trusel et al., 2013).

Thank you for reminding us of this useful reference, which we will refer too when stating the melt rates of the northern GVIIS.

Line 90: Reynolds (1981) discusses observations of moulins on George VI, so consider modifying this sentence.

Thank you for pointing this out. We will modify this sentence to state that moulins have been observed in the pressure ridges near to the margins of GVIIS by Reynolds (1981).

Line 91: Quantify 'high' basal melt rates and thinning rates.

Basal melt rates are < 6 m yr$^{-1}$ in the southern GVIIS (Adusumilli et al., 2020), and thinning rates have been < 2 m y$^{-1}$ in this area (Pritchard et al., 2012). We will state these statistics (and references) in our revised manuscript.

Line 93: Consider adding one line in this paragraph quantifying ice flow speeds on northern GVIIS, since ponding preferentially occurs on slower-moving ice.

This is a good suggestion and we will add the following sentence to this section: "*The ice shelf decelerates as it flows westwards across the sound, with ice velocities on the northern GVIIS varying from ~ 400 m yr$^{-1}$ near the grounding line to ~ 30 m yr$^{-1}$ near Alexander Island (Bishop and Walton 1981).*"

Line 95: I suggest also citing Lucchitta and Rosanova (1998) here as well.

Thank you, we will add this reference.

Line 99: I suggest making it clearer in this paragraph that these three types of surface lakes form every austral summer to make it clearer you are not just referring to 2019/2020. In addition, consider adding sub-panels to Figure 1 to show examples of these three types, or a separate Supplementary Figure.

This is a good point; we will re-word this sentence to read "*On the northern GVIIS, three types of surface lake patterns tend to form each summer*". However, we currently do not plan to add an additional figure to our paper to highlight these three lake types. This is because although we observe these three lake types in 2019/2020, our study does not advance scientific knowledge about these lakes, which have previously been described in detail in the literature. For example, the first two lake types are described in detail in Reynolds (1981) alongside photos and diagrams (their figs 3a and 5), and the third lake type is the focus of the study by LaBarbera and MacAyeal (2011). These references are already noted in our paper.

Line 100: I suggest you also cite Hambrey and Dowdeswell (1994) here, and perhaps add to the end of this sentence that ice flowlines are surface manifestation of longitudinal foliation.

We will additionally cite Hambrey and Dowdeswell (1994) here, and add the following text to the end the current sentence "*….similar to the dominant pattern of lakes observed on the Amery Ice Shelf (Hambrey and Dowdeswell, 1994).*"  However, we are not sure we fully understand the reviewers suggestion about stating that the "ice flowlines are the surface manifestation of longitudinal foliation", as

to us, 'ice flowlines" are simply the directions along which ice flows across the ice shelf, not only at the surface, but throughout the depth of ice.

Line 104: Please consider also citing Langley et al. (2016) and Arthur et al. (2020b) here, which record observations of down-ice lake advection on Langhovde Glacier and Shackleton Ice Shelf, East Antarctica.

We will add these references.

Line 111: I suggest continuing the final sentence with: 'as described above, enabling it to support a large surface area of surface meltwater (Alley et al., 2018).

Thank for this good suggestion. We will add this extra text and reference.

Line 117: Be consistent with the use of northern/north GVIIS; I think northern is used most frequently throughout.

Previously we had used 'on north GVIIS' when trying to be brief (e.g. in sub headings, and figure captions), whereas we had used 'on the northern GVIIS' in the main paper text. However, we are very happy to use 'northern' (not 'north') throughout our revised paper.

Line 125: What is the uncertainty associated with this microwave brightness temperature product and the ASCAT product in Section 3.2? There could be more discussion of this.

Regarding uncertainties with the microwave radiometer and scatterometer products, please see our response to this reviewer's general comment on page 2 of this response letter.

Line 129: I suggest adding one sentence after this along the lines of: the sensor measures the emitted energy from the surface and sub-surface, proportional to the brightness temperature (which increases with the presence of liquid water and which increases absorption and emissivity).

We thank the reviewer for this helpful suggestion, though we will modify their suggested sentence slightly to avoid overlap with the previous sentence, and to insert the following two sentences at the beginning of the paragraph: "*Microwave radiometer observations of melt, expressed as brightness temperatures, depend primarily on the snow temperature profile and emissivity (Zwally, 1977). When liquid water exists in the snow, there is a significant increase in the absorption, and therefore an increase in the microwave emissivity, resulting in higher brightness temperature.*"

Line 142: Briefly state why grid cells > 1700 m a.s.l. were masked out, presumably to only show data corresponding to ice shelf areas.

The reviewer is correct with their assumption. Cells > 1700 m a.s.l. were masked out such that only melt on ice shelves was predominantly analyzed. We will reword this sentence to read as follows: "*Grid cells with surface ice elevations > 1700 m a.s.l. were masked out so that melt over the ice shelves was predominantly analysed and to avoid large topographic features in the radiometers field-of-view (i.e., mountain peaks).*"

Line 148: Consider adding a citation here?

We will add a reference to Zwally (1977) here.

Line 161: I suggest adding a sentence explaining where this threshold comes from i.e. was based on empirical comparisons with QuikSCAT-derived melt (Ashcraft and Long, 2006).

We will add the following text to the end of this sentence: "*…..ASCAT signal is lower than the winter mean signal minus 3 dB, as proposed by Ashcraft and Long (2006) using a melt model and QuikSCAT Ku-band (13.4 GHz) observations*".

Line 166: Add 'scatterometer-derived' before 'cumulative melt days'.

We will add this.

Line 177: Perhaps briefly outline here the threshold-based algorithms other than NDWI that this method uses, i.e. NDSI and others, or else list which bands are used.

Adding details of all the bands and band combinations that we used to the manuscript would be extremely lengthy (at a minimum this would be two additional paragraphs, i.e. one paragraph per sensor, with 5 – 6 sentences in each paragraph), and given all this detail is in Moussavi et al (2020), which we follow exactly, we not think repeating this information in our paper is of value. We will, however, add a little more detail to the current text, and will re-word the section to read: "*Moussavi et al's (2020) method, developed separately for Landsat 8 and Sentinel-2, combines separate threshold-based algorithms to detect (1) lakes, (2) rocks, and (3) clouds. Optimal thresholds for each band and band combination (e.g. Normalized Difference Water Index (NDWI), Normalized Difference Snow Index (NDSI) and others) were determined by creating a training dataset based on selected Landsat 8 and Sentinel-2 images, which represented spectral properties of several classes (e.g. lakes, slush, snow, clouds, rocks, cloud shadows). Most notably, to classify liquid water-covered pixels, the NDWI is used (Pope et al., 2016; Bell et al., 2017), with NDWI thresholds of > 0.19 and > 0.18 for Landsat 8 and Sentinel-2, respectively*".

Line 182: Consider also citing Bell et al. (2017) and Arthur et al. (2020b) here.

We will add these references here.

Line 184: This depth-reflectance algorithm makes a number of assumptions which I think are worth briefly outlining in an additional sentence, including the assumption of homogenous lake bottom albedos, minimal wind-driven light scattering, etc (Sneed and Hamilton, 2007).

Please see our earlier response to a similar comment made by this reviewer on page 4 of this letter.

Line 185: There is no discussion currently in this section of false positives. How did you deal with these (if there were any), and were they manually removed?

The major issue with any image classification method, including ours, is false positives, which results in overestimation of lake areas. The only way to reduce these errors would be to post-process products, i.e. manual inspection of results, which is very labor intensive and is rather subjective. Therefore, we did not manually post process the images used in our study.

Line 218: I think 'warm' here more accurately describes foehn winds than 'hot'. Also suggest adding at the end of this sentence 'and commonly occur on the AP', and citing Luckman et al. (2014).

We agree re. foehn winds being 'warm' (not hot) and will change this. We will also add these additional words and reference.

Line 219: I suggest adding an indication here of how steep the topography is, e.g. maximum slope.

Having shown that foehn winds are not a dominant contributor to surface melt during this 2019/20 season, we think that the further examination of drivers for foehn winds (including the slope of Alexander Island on the NW margin of the GVIIS) to be beyond the scope of our current study.

Line 339: This is an interesting finding. I wonder whether it would be worth adding an additional sentence (either here or at the end of this section) explicitly summarising these observations demonstrate you can still get record melt when conditions are generally warmer, with no involvement of foehn wind events.

This is a useful point, and we will add the following sentence to the Conclusions to explain this: "*It is therefore notable that although the high melt event over the northern GVIIS is 2019/2020 was caused by warmer than average air temperatures, such local weather conditions were not foehn-driven*". Also, we will add additional sentences to the Abstract and Results to emphasize that the high melt in 2019/2020 was not foehn driven.

Line 393: Are there any measurements of firn air content/thickness on George VI?

There are a couple of studies that have modelled firn air content (FAC) over GVIIS (e.g Ligtenburg et al., 2014), but the spatial resolution of those studies is insufficient for us to be able to state differences in FAC between the north and south GVIIS.

Line 410: I don't think you can necessarily suggest that surface melt volumes were highest in 2019/2020 out of the prior 31 seasons without having explicitly derived volume estimates from imagery pre-2013.

We thank the reviewer for their comment about this, and having thought about this paragraph some more, we agree that it is best deleted from our revised paper.

Line 419: Change 'zero re-freeze' to 'no refreezing occurred'.

We will change this.

Line 434: Explain 'shoulder seasons' – do you mean colder seasons (autumn/winter?).

By "shoulder seasons" we meant all seasons excluding summer, but mainly winter. However, we now realize that our terminology is not clear, and we apologize for that. We will reword the sentence to read: "*This is predictable foehn flow behaviour, e.g., over the Larsen C, foehn winds are strongest in winter, when wind speeds are generally higher (Datta et al., 2019; Wiesenneker et al., 2018).*"

Line 446: Change 'back to 2013' to 'from 2013 to 2020' and suggest adding to the last part of the sentence: 'was also exceptional in areal extent and volume [..]'.

We will make these changes.

Line 454: Add AOI area here in brackets to remind readers of its size in comparison with Larsen B in the next sentence. Also, what was the maximum lake depth?

This is a good suggestion and we will add the area of the AOI (7850 $km^2$) in brackets. However we are not convinced that stating the maximum depth here in the Conclusions is suitable given this value appears to be rather an outlier. Instead, we will state this value in the Results.

Line 458: I suggest also citing Alley et al. (2018) here.

We will add this reference.

Line 460: Consider showing in a Supplementary Figure the lakes mapped in this study overlaid onto the areas classed as vulnerable to hydrofracture by Lai et al. (2020).

We thank the reviewer for this suggestion, which we will consider. Our main reservation for doing what they suggest is that we think such a figure would be too 'busy' to be easily interpretable, and therefore it would not serve it's purpose well. We could instead insert just a map of the vulnerable areas of the GVIIS from Lai et al. (2020), but in which case, it makes more sense to simply refer the readers directly to the figure in that study.

Figure S5: I notice there are two particularly deep lakes – why, out of interest, do you suggest this is?

We think that these areas may remain as topographic depressions from season to season due to the flow regime in this area. Additionally, their basin depths may have been enhanced due to enhanced lake bottom ablation in these areas. In other words, the greater the number of days that meltwater ponds in a specific area, the greater the effect of enhanced melting at the lake bottom due to the lower albedo of the lake water compared to the surrounding water. However, as our study focusses on the total volumes of surface meltwater each season, rather than behaviours of specific lakes, we suggest it is beyond the scope of the current study to discuss the evolution of these deep lakes in the paper.

Line 31: minor point, but check here and throughout that citations are listed chronologically.

We will do this.

Line 49: Add comma after 'However'.

We will do this.

Line 100: 'most extensive' rather than 'the largest'?

We will change this.

Line 106: Change en-echelon to 'en-echélon' and consider adding a brief description for those unfamiliar with this term, e.g. closely spaced, sub-parallel.

Thank you, we will make both of these changes.

Line 120: Change '(from 2013)' to 'from 2013-2020' and the same with '(from 2017)'.

*We will change this.*

Line 125: Insert 'passive' before 'microwave radiometer'.

*As a radiometer is, by definition, a passive sensor, we have instead reworded this part of the sentence to read "… was derived from microwave radiometer (i.e. passive) observations…".*

Line 132: Hyphenate 25 km.

*We will do this.*

Line 157: Hyphenate 4.45 km and consider briefly outlining the SIR algorithm.

*We will reword this sentence to read as follows: "The 4.45-km enhanced product is obtained by applying the Scatterometer Image Reconstruction (SIR) algorithm with filtering (Lindsley and Long, 2016), which is used to improve the spatial resolution of irregularly and oversampled data (Early and Long, 2001)."*

Line 159: Hyphenate 'SMMR-based'.

*We will do this.*

Line 173: Rephrase to 'to selected multispectral imagery (see paragraph below)'.

*We will do this.*

Line 197: Are italics needed in this paragraph?

*We are happy to remove these italics.*

Line 216: Remove 'periods'.

*We will do this.*

Line 220: citation should be Wiesenneker et al., 2018 (full reference is correct).

*We will correct this, however we note that the following spelling is correct: "Wiesenekker" (not Wiesenneker).*

Line 291: remove commas after 2020 and AOI.

*We will do this.*

Line 327: Change 'refreeze' to 'refreezing'

*We will change this.*

Line 390: Re-word sentence to 'meltwater ponding is not observed in the optical imagery until mid-December'.

We will change this.

Line 495: Please add volume, issue and page numbers: 44(6), 837-869.

We will do this.

Additional references that we will add

Alley KE, Scambos TA, Anderson RS, et al. Quantifying vulnerability of Antarctic ice shelves to hydrofracture using microwave scattering properties. Remote Sensing of Environment 210: 297–306, https://doi.org/10.1016/j.rse.2018.03.025, 2018.

Arthur, J. F., Stokes, C. R., Jamieson, S. S. R., Carr, J. R., and Leeson, A. A.: Distribution and seasonal evolution of supraglacial lakes on Shackleton Ice Shelf, East Antarctica, The Cryosphere Discuss., https://doi.org/10.5194/tc-2020-101, 2020b.

Langley ES, Leeson A. A., Stokes. C. R., et al. Seasonal evolution of supraglacial lakes on an East Antarctic outlet glacier. Geophysical Research Letters 43(16): 8563–8571, 2016.

Lucchitta, B. K. and Rosanova, C. E. Retreat of northern margins of George VI and Wilkins Ice Shelves, Antarctic Peninsula. Annals of Glaciology 27: 41–46, 1998.

Luckman A, Elvidge A, Jansen D, et al. Surface melt and ponding on Larsen C Ice Shelf and the impact of fohn winds. Antarctic Science 26(6): 625–635, 2014.

Trusel, L. D., K. E. Frey, S. B. Das, P. Kuipers Munneke, and M. R. van den Broeke, Satellite-based estimates of Antarctic surface meltwater fluxes, Geophys. Res. Lett., 40, 6148–6153, doi:10.1002/2013GL058138, 2013.

---

## Author Comment (AC2) · 8 Jan 2021

**Authors' response to anonymous Reviewer #2**

This is an engaging study investigating the 2019/2020 austral melt season on the north George VI Ice Shelf, Antarctic Peninsula. The authors combine different datasets such as satellite microwave radiometer/scatterometer data, optical satellite imagery and local meteorological station data. The authors highlighted exceptional melt and surface ponding conditions in 2019/2020, and they concluded that warm temperatures were likely triggered by warm northwesterly and northeasterly low-speed winds, rather than Foehn winds.

The paper is well written, and the results are supported by the data and methods. The study has the potential to become a fully relevant article to concept of The Cryosphere and my criticisms are with regards presentation rather than science.

We thank this reviewer for their very useful remarks.

Specific comments:

Lines 31-32: Could you please specify the rate of mass loss?

We will re-word this sentence to read: "*The rate of mass loss from the AP has tripled since the 1990s, with an average of 24 Gt yr-1 from 1979 to 2017 and an acceleration of 16 Gt yr$^{-1}$ per decade (Rignot et al., 2019).*" We also note, that as these statistics are currently noted in the Section 2 of our paper ("Study Site"), we will delete them from that section in our revised paper.

Lines 53-54: Is there any projected estimation of contribution of the AP ice shelf melting to the global sea level rise? If yes, that information would be useful.

We will add information about the potential sea level contribution from the George VI versus Larsen C ice shelves. However, we suggest it will be more appropriate to add those details at the end of Section 2 ("Study Site"). The sentences we will add are as follows: "*Due to the strong buttressing forces that the GVIIS provides relative to the large volume of grounded ice in Palmer Land, if this ice shelf were to completely collapse, the resultant acceleration of the inland glaciers would add < 8 mm to global sea levels by 2100 and < 22 mm by 2300 (Schannwell et al., 2018). In contrast, sea level contributions resulting from the collapse of the much larger Larsen C Ice Shelf would be relatively low (< 2.5 mm by 2100, < 4.2 mm by 2300).*"

Lines 56-58: I would add a link of official WMO status on this evaluation: https://public.wmo.int/en/media/news/new-record-antarctic-continent-reported. In addition, to my knowledge, WMO is evaluation the value of 18.4◦C., not the 20.75◦C. There has been no official effort/information to evaluate the value of 20.75◦C. Further- more, this value was not recorded at the official Marambio station. I recommend the authors to make sure that whether this temperature is being evaluated or not. If not, I recommend them to remove the statement in lines 58-60.

Thank you, we will add the weblink. We also agree that this 20.75°C record does not now appear to be in evaluation by the WMO, so we will delete this sentence (in addition to removing the following sentence about the prior record on Signy Island, which is also no longer relevant).

Lines 60-63: Could you please quantify melt amount in this part too? Because, you are comparing GVIIS with the other AP ice shelves. Some numerical values would help the reader to see the differences.

Quantifying melt volumes on ice shelves in addition to the GVIIS is beyond the scope of this study, however we will add an additional sentence to this section that will qualitatively describe and compare meltwater ponding on AP ice shelves that is observed in the MODIS mosaic shown in Figure 1a.

Lines 87-94: What makes the northern GVIIS more vulnerable to high surface summer melt rates? I would expect to consider rates of basal melting, nonetheless, you state that rates of basal melting are greatest at the southern end of the GVIIS. Therefore, a clarification might be needed to have a better idea on different physical processes at the northern and southern ends.

The northern GVIIS experiences warmer air tthan the southern GVIIS (e.g. see Datta et al., 2018). Additionally, precipitation rates in the North are lower than in the south, which is at least partly attributable to the precipitation shadow of Alexander Island. We will reword the first sentence of this paragraph to read the following, which hopefully clarifies the key differences in the surface processes over the GVIIS: "*Compared to the southern GVIIS (~72$^o$00' S to 77$^o$00), the northern GVIIS (~70$^o$30' S to ~72$^o$00' S) experiences higher surface summer melt rates (< 250 mm w.e. yr-1; Trusel et al., 2013; Datta et al., 2018) and lower accumulation rates (< 0.2 Mg m-2 yr-1; Bishop and Walton, 1981; Reynolds, 1981); the latter is attributed to the presence of a precipitation shadow down-wind of Alexander Island (Bishop and Walton, 1981).*"

Regarding basal melting, we are not sure what this reviewer means by "I would expect to consider rates of basal melting", as we do already give detailed information about the spatial variation in basal melt rates, and the reasons for high basal melt rates, in the current version of our paper (current lines 91 – 94). However in our revised paper, we will also state basal melt rates from Adusumilli et al. (2020), as requested by Reviewer 1.

Lines 123, 208-212: Is there any elevation difference between the AWS station and ice shelf?

This is a good question. The AWS elevation is 66 m, which is similar to the ice shelf surface elevation in this region (50 – 60 m, given that the ice shelf thickness is 500 – 600 m). We will state the AWS's elevation along the AWS's coordinates in Section 3.4.

Line 213: What does it mean exactly "using 12-hour data" for the 1999-2020 period? I understand that you use only one time step as a daily mean temperature for the 1979- 1999 period. If yes, what is the exact hour of the observation (morning, late afternoon)?

We apologize for being unclear. By "12-hour data", we are referring to data that were recorded 12 hours apart; at noon and midnight. Therefore each daily mean value from 1979 to 2020 (NB. not just 1999 to 2020 as this reviewer states) is calculated from two data points. We will add all these details to the manuscript.

For the second period, did you pick up the same hour with the 1999-2020 period or did you take a temporal mean of 12-hour data?

We perform high temporal resolution analysis is from 2007 to 2020, which is (as it currently stated in the manuscript) when "*when AWS data are at a high frequency (10 minute intervals) and data gaps are minimal*". Therefore, for this period, we use the 10 minute data (not the 12-hour data) to calculate the length of time when temperatures are continuously at/above 0$^o$C. We will ensure we make this clearer in our paper.

Line 223: Could you please specify why there exist lower mean speeds over north GVIIS?

Although the study by van Wessem et al (2015; their Figs 10 and 11) show that in general, wind speeds over North GVIIS appear to be similar to those in the Scar Inlet/Larsen C region, foehn winds will likely be slower in speed at Fossil Bluff compared to the Cabinet Inlet due to the lower mean elevation of Alexander Island's topography compared to the AP mountains to the west of the Scar Inlet (e.g. see van Wessem et al (2015); their Fig 1). Therefore, we will edit the current sentence in the manuscript to read: *"We use a wind speed threshold of 1.5 ms$^{-1}$ instead of the higher threshold of 3.5 ms$^{-1}$ used by Datta et al. (2019) for the Cabinet Inlet AWS to account for lower foehn wind speeds over north GVIIS, which result from the lower mean elevation of the mountains on Alexander Island compared to those on the AP west of Cabinet Inlet (van Wessem et a., 2015)."*

We also note that our wind speed threshold of 1.5 ms$^{-1}$ is actually more liberal than the 3.5 ms$^{-1}$ threshold used by Datta et al (2019). In other words, although we only calculated 9 hours of foehn conditions in the 2019/2020 melt season, if we had used the higher Datta et al. threshold, we would have calculated even fewer hours of foehn conditions for the northern GVIIS.

Lines 238-239: In your analysis, I understand that Larsen C shows a record high melt year in 1992-1993, not in 2019-2020 on the contrary to the findings of Bevan et al. Is that related just with the different datasets? Can you please specify potential uncertainties of each product?

We will add the following sentence to our revised manuscript, which we hope will more clearly describe the discrepancy: *"This finding is contrary to the results of Bevan et al. (2020), who report that Larsen C experienced a 41-year record high melt year in 2019/2020. Bevan et al's (2020) results are based on microwave radiometer (SMMR/SSMI) data for melt seasons from 1979/1980 until 2016/2017, followed by microwave scatterometer (ASCAT) data from 2017/2018 to 2019/2020. In contrast, we use SSMR/SSMI data over the AP for the full 1979 to 2020 period to preserve consistency. As we explain in Section 3.2, ASCAT C-band radar is likely to be more sensitive to melt at depth than microwave radiometers, thus resulting in Bevan et al's (2020) higher calculated melt over Larsen C in the 2019/2020 season after combining different data sources into one time series."*

Regarding the potential uncertainties of each of the melt products, this has now been addressed in response to one of Reviewer 1's main comments, so please refer to our long reply there.

Lines 326-334: Why not to compare the longest warm periods with the volume changes? For instance, are the longest periods coincide with the largest volume change?

We thank the reviewer for this useful idea, and we will insert the following sentence into the Results section (4.3): *"In 2019/2020, volumes of meltwater ponding are highest in early January and then again in early February; corresponding with periods when air temperatures are ≥ 0$^{o}$C for extended periods."*

Additionally, we will insert the following sentence into the Discussion section (i.e. '5.3. Local climatic controls of the 2019/2020 melt event'): *"For example, we note that sustained warm temperatures in early January (Fig. 7b, periods A and B) and early February (Fig. 7b, periods C and D) coincide with periods when surface meltwater volumes derived from optical imagery are relatively high (Fig. 5b)."*

Lines 335-339: I think the authors should also discuss the potential role of the warm air advection given the low foehn conditions. I suspect that regional sensible heat transport seems to be one of the main contributors of the 2019/2020 melt season, particularly for the first two weeks of February.

We completely agree that warm air advection likely drove the sustained high air temperatures in 2019/2020, and therefore the high melt in this season. However, we are wondering if the reviewer wrote

this comment before reading the following two paragraphs in our paper (current lines 349 – 359), which describe the results of our analysis of wind speed alongside air temperature data (i.e. warm air advection). In particular, our current manuscript states: "*in the 2019/2020 melt season, we show that both northwesterly and northeasterly winds show warmer temperatures at lower wind speeds*" (current lines 358/359). Melting due to the increase of warm, low-speed winds from the NE and NW is also discussed in the Discussion, and mentioned in the Conclusion and Abstract. In our revised manuscript, we will ensure that we clarify that what we are describing is indeed warm air advection. For example, we will add the following sentence to the beginning of the paragraph starting on current line 349: "*Finally, we analyse the potential role of warm air advection, resulting in sensible heat transport, on the high melt in 2019/2020.*"

Lines 349-359: Can you specify where the calculated 9 hour of foehn is located in this time series (Fig. 6)?

The time periods when foehn conditions are present are already described in the previous paragraph that specifically focusses on foehn winds (current lines 335 – 337), and these periods are already labelled in Fig 6b (blue circles), so we are wondering if the reviewer missed this. The relevant sentence currently reads: "*Foehn conditions (as described in Section 3.4) are only present for about 9 hours over the entire 2019/2020 season (Fig 5, blue line), and occur in early and late summer (Fig. 6b, blue circles) when winds are typically stronger.*"

Lines 431-438: I am a bit disconnected here. So, I understand that rather than local conditions (i.e., Foehn winds) one should expect to see regional- and/or large-scale warm advection. In this case, I would expect to see similar warming rates over the GVIIS and Larsen C. However, in Fig. 2c there are notable warming differences be- tween these two regions. Do the authors have any idea for these differences?

We would first like to clarify to this reviewer that Figure 2 shows surface melt, not warming (though surface melt is obviously driven by the surface energy balance). Second, we agree that this figure shows significantly higher melt over the George VI and Wilkins ice shelves compared to the NE areas of the AP (inc. Larsen C) in 2019/2020, and we will add the following extra sentence to Results section 4.1 to make this fact clearer: "*For the AP, cumulative melt days in the 2019/2020 austral melt season are highest in the southwestern areas of the AP (including the Wilkins and George VI ice shelves), in addition to the northern area of the Larsen C Ice Shelf (Fig. 2b). In comparison, cumulative melt days in 2019/2020 are relatively low over the southern areas of the Larsen C.*" However, as the main aim of this study is to focus on melt on the northern GVIIS, and the local atmospheric controls of this melt, it is outside of the present study's scope to perform modelling to establish why the SW area of the AP experienced more melt than the NW area of the AP in 2019/2020.

Fig. 1 and Fig. 2: Could you please use a larger font size for the lat/lon coordinates?

We will do as suggested for Figs 1 and 2, as well as for Fig S1.

Overall suggestion: As you use different datasets with different spatial and temporal resolutions, I would add a table for the information given in data and methods (e.g., name of the product, resolution, temporal coverage etc.). This makes it easier for reader to follow the result.

As we actually only use four data products in total, i.e. two microwave products (SMMR/SSMI and ASCAT) and two optical image products (Landsat 8 and Sentinel-2), we do not think that a table of details about these products will be sufficiently beneficial, especially as the details about each product

are clearly stated at the beginning of each section in the Methods (i.e. SMMR/SSMI is described in Section 3.1, ASCAT is described in Section 3.2, and Landsat 8/Sentinel-2 are described in Section 3.3). However, we are happy to include such a table if the Editor would like us to.

Additional references that we will add

Bishop, J. F. and Walton J. L. W. Bottom melting under George VI Ice Shelf, Antarctica. Journal of Glaciology. 27(97), 429-447, 1981

Datta, R. T., Tedesco, M., Agosta, C., Fettweis, X., Kuipers Munneke, P., & van den Broeke, M. R. Melting over the northeast Antarctic peninsula (1999–2009): Evaluation of a high-resolution regional climate model. The Cryosphere, 12(9), 2901–2922. https://doi. org/10.5194/tc-12-2901-2018, 2018.

Schannwell, C., Cornford, S., Pollard, D., and Barrand, N. E.: Dynamic response of Antarctic Peninsula Ice Sheet to potential collapse of Larsen C and George VI ice shelves, The Cryosphere, 12, 2307–2326, https://doi.org/10.5194/tc-12-2307-2018, 2018.

---

## Author Response (AR1)

Dr Alison Banwell
Research Scientist
CIRES
University of Colorado Boulder
USA

email: alison.banwell@colorado.edu

11 January 2021

Dear Dr. Howell,

We have received three reviews of our manuscript entitled '32-year record-high surface melt in 2019/2020 on the northern George VI Ice Shelf, Antarctic Peninsula'. Overall, all three reviews were supportive of the paper and its relevance to the wider research community.

All of the reviewer's comments have been extremely helpful in further improving the manuscript. The specific changes made can be found in our responses below. Referee comments are in black, and our replies are in blue.

Please also find attached a revised manuscript with all tracked changes. Please note that the line numbers referred to in the text below are those on the 'tracked changes' manuscript.

Thank you for taking your time to consider our revised manuscript, we hope it is now acceptable for publication.

Sincerely,

Alison Banwell, on behalf of all co-authors

**Authors' response to Reviewers**

**Reviewer #1**

This manuscript presents a quantitative analysis of surface meltwater on the George VI Ice Shelf, Antarctic Peninsula, focusing on the most recent melt season (2019/2020) and setting this in the longer-term record of melt. The authors use data from a number of sources, including microwave radiometer and scatterometer, automatic weather station measurements and a previously published algorithm to classify surface meltwater ponding from Landsat 8 and Sentinel-2 imagery.

Meltwater has been linked to the instability and collapse of Antarctic Peninsula ice shelves. Although the focus and methods of this study are not novel, it quantifies the recent anomalous melt event on this ice shelf, which is important given that such record-high melt events are set to become more frequent in future. Therefore, it is my view that the findings from this manuscript are of broad interest to the cryospheric community.

In general, I would like to complement the authors on their well-written and clearly-structured manuscript, and the study rationale and methods are well-justified. The results build upon previous work that has reported surface meltwater lakes on this ice shelf by providing a time series of ponded surface meltwater together and analysing this alongside microwave observations of melt together with local climatic controls.

We thank this reviewer for their complementary remarks about our paper and we are pleased to hear that they think it will be of broad interest to the cryospheric community.

The authors use optical Landsat 8 and Sentinel-2 imagery to derive ponded meltwater volumes. However, I wonder why historical satellite imagery pre-2013 was not used to supplement this record and set 2019/2020 within the longer-term context of surface ponding? I suspect this may be related to difficulties applying the lake depth radiative transfer model to historical imagery, but it would be worth justifying.

Analysis of pre-2013 imagery would require tuning Moussavi et al's threshold-based algorithm for the Landsat 7 sensor (and for prior Landsat sensors). This could be done, however, with much of Landsat 7 data missing (due to SLC off gaps), lake volumes derived from this sensor would not be easily comparable to L8 and S2. In other words, the inclusion of Landsat 7 data will not necessarily extend the time series record, simply because its data is not comparable to Landsat 8 and Sentinel-2. We briefly state these reasons in our revised paper (lines 264 - 268).

There could be more of a discussion on the fate of surface meltwater during and at the end of the melt season. For example, is the decrease in meltwater volume in mid-late January associated with refreezing, or is there any evidence of englacial lake drainage? Similarly, do the authors observe any rapid drainage events, and if so at what point in the melt season?

This is a helpful suggestion and we have included some additional statistics in the main text (inc. mean depth and maximum depth of water-covered pixels, see lines 466-467). We have also added some sentences that describe links between the total volume of surface ponding and air temperatures, particularly for the 2019/2020 season (lines 474-476). However, to intercompare volumes of meltwater between different austral melt seasons, the optical image analysis part of this study focusses on temporal variations in the total area and volume of surface meltwater (and particularly the maximum values) each season. Therefore, we did not produce timeseries of data for individual lakes, and

producing those data would involve re-running the algorithm and performing a substantial new amount of analysis that is beyond the scope of the current study. However, from visual analysis of the key images, we have not noticed any evidence of rapid lake drainage events, and due to the compressive flow regime (see line 106), we also would not expect such events to be common.

There is also a lack of discussion of uncertainties, especially melt detection uncertainty using the microwave brightness temperature product and the ASCAT product.

Addressing uncertainties associated with microwave data products of binary melt/no melt information is challenging. In-situ measurements (e.g., of snow temperature, liquid water content) for true validation purposes are rare. Assessment against air temperature measurements have been done in the past. In our study, two distinct microwave remote sensing techniques and algorithms were used to bring confidence in our conclusion; we used both microwave radiometer (SMMR/SSMI) data and microwave scatterometer (ASCAT) data. Although the figure showing the ASCAT-derived results was previously in the Supplementary Information, we have now moved it into the main paper (it is now Fig 4).

Additionally, to further address the reviewer's comment and to bring new content to our study, we have analyzed the sensitivity of the melt detection algorithms to their threshold values; the most uncertain part of the algorithms. Please see below for results of this analysis for both melt products, which confirm that our conclusion of the exceptional 2019/20 melt year is robust.

SMMR/SSMI uncertainty analysis

The figure below shows the brightness temperature temporal evolution for one pixel in North GVIIS (lat, lon = -72.287, -67.579) over the course of the 2019/2020 melt season. Three algorithm thresholds, defined by the mean brightness temperature plus the standard deviation multiplied by a coefficient, are also indicated. The coefficient of 3 is nominal (Torinesi et al. 2003) and was used our study, as in other past studies. For this single pixel, the resulting cumulative melt days for each threshold are: coef: 2.5 -> 107, coef: 3 -> 103, coef: 3.5 -> 99.

[Figure]

Using each of these three coefficients (2.5, 3, 3.5), we have also calculated the spatially-averaged cumulative melt days over the northern GVIIS for all melt seasons from 1979/1980 to 2019/2020 (apart from 1987/1988 due to missing data), the results for which are shown in the bar plot below. We estimate that changing the coefficient from 2.5 to 3.5 produces a change in the cumulative melt days in

the order of 10%. Importantly, as shown by the plot below, the 2019/2020 melt season is exceptional regardless of the choice of coefficient used in the SMMR/SSMI analysis.

[Figure]

ASCAT uncertainty analysis

The bar plot below shows the ASCAT-derived spatially-averaged cumulative melt days over northern GVIIS for melt seasons from 2007/2008 to 2019/2020 using three thresholds; 2 dB, 3 dB (the value we use in our study), and 4 dB. As this plot shows, varying the threshold does not alter the conclusion that the 2019/2020 melt season experienced the highest cumulative melt days.

[Figure]

The revised manuscript includes the following sentences in Results section 4.2: "*Since addressing uncertainties associated with microwave data products of binary melt/no melt information is challenging, this study uses two distinct microwave remote sensing techniques and algorithms to build further confidence in our conclusion. Moreover, our analysis of the sensitivity of the microwave radiometer (Fig. S4) and scatterometer (Fig. S5) melt detection algorithms to decreasing/increasing their threshold values shows that the 2019/2020 melt season remains exceptional, and a 32-year record.*" Figs. S4 and S5 refer to the two bar plots above, which we have inserted into the Supplementary Information for our revised paper.

In addition, the reader should be made aware of the limitations in using the depth retrieval algorithm applied to optical satellite imagery (see Sneed and Hamilton, 2011 and Pope et al., 2016).

Regarding limitations associated with using the depth retrieval algorithm applied to optical images, in our revised manuscript, we have added the following sentence about the assumptions that the depth algorithm makes: "*This approach makes a number of assumptions, including that the lake bottom has a homogenous albedo, there is little to no particulate matter in the water column to alter its optical properties, and that there is minimal wind-induced surface roughness (Sneed and Hamilton, 2007).*"

I think it could also be highlighted more clearly in the manuscript that this record surface melt in 2019/2020 was unrelated to foehn-driven melting (see specific comments).

We agree with this comment and we have added additional sentences to clarify this in the Abstract (line 26), Results (line 541) and Conclusion (line 755) sections.

Once the authors address these issues and my specific comments below, I can therefore recommend that this manuscript is suitable for publication in The Cryosphere.

Specific comments:

Line 25: consider quantifying 'low-speed' winds here, i.e. ≤ 7.5 ms$^{-1}$

We have done this.

Line 63: Consider adding either an additional sentence here or an additional panel in Figure 1 showing which other ice shelves experienced increased meltwater ponding in 2019/2020 to provide further context. Although it is mentioned in the Figure caption that Larsen C, Wilkins and Bach also experienced ponding, this is not immediately clear from Panel A.

We thank the reviewer for these suggestions. We would prefer not to add an additional panel to figure 1 as that would mean that panel (b) would need to become smaller, meaning lakes on GVIIS would be less visible. Instead have added the following sentence to the paper: "*However, as Fig. 1a shows, in 2019/2020, surface meltwater ponding was also prevalent on the northwestern Larsen C (Bevan et al., 2020), the eastern Wilkins (also visible in the bottom left corner of Fig 1b), and the northern and northwestern Bach ice shelves.*"

Line 66: I suggest either enlarging the latitude-longitude labels on Panel A, or adding them to Panel B. In addition, I think it would be helpful to add an arrow labelling ice flow direction.

We have enlarged the latitude-longitude labels on panel a), and we have also added an arrow to indicate ice flow on panel b).

Line 87: Quantify surface summer melt rates (e.g. up to ~400 mm w.e. yr, Trusel et al., 2013).

Thank you for reminding us of this useful reference, which now refer too when stating the melt rates (line 111).

Line 90: Reynolds (1981) discusses observations of moulins on George VI, so consider modifying this sentence.

Thank you for pointing this out. We have modified this sentence to state that moulins have been observed in the pressure ridges near to the margins of GVIIS by Reynolds (1981).

Line 91: Quantify 'high' basal melt rates and thinning rates.

Basal melt rates are < 6 m yr$^{-1}$ in the southern GVIIS (Adusumilli et al., 2020), and thinning rates have been < 2 m y$^{-1}$ in this area (Pritchard et al., 2012). We have stated these statistics (with references) in our revised manuscript.

Line 93: Consider adding one line in this paragraph quantifying ice flow speeds on northern GVIIS, since ponding preferentially occurs on slower-moving ice.

This is a good suggestion and we have added the following sentence to this section (lines 106 – 108): "*The ice shelf decelerates as it flows westwards across the sound, with ice velocities on the northern GVIIS varying from ~ 400 m yr$^{-1}$ near the grounding line to ~ 30 m yr$^{-1}$ near Alexander Island (Bishop and Walton 1981).*"

Line 95: I suggest also citing Lucchitta and Rosanova (1998) here as well.

Thank you, we have added this reference.

Line 99: I suggest making it clearer in this paragraph that these three types of surface lakes form every austral summer to make it clearer you are not just referring to 2019/2020. In addition, consider adding sub-panels to Figure 1 to show examples of these three types, or a separate Supplementary Figure.

This is a good point; we have re-written this sentence to read "*On the northern GVIIS, three types of surface lake patterns tend to form each summer*". However, we have decided not to add additional figure to our paper to highlight these three lake types. This is because although we observe these three lake types in 2019/2020, our study does not advance scientific knowledge about these lakes, which have previously been described in detail in the literature. For example, the first two lake types are described in detail in Reynolds (1981) alongside photos and diagrams (their figs 3a and 5), and the third lake type is the focus of the study by LaBarbera and MacAyeal (2011). These references are included in our paper.

Line 100: I suggest you also cite Hambrey and Dowdeswell (1994) here, and perhaps add to the end of this sentence that ice flowlines are surface manifestation of longitudinal foliation.

We have added the following text to the end the current sentence "*….similar to the dominant pattern of lakes observed on the Amery Ice Shelf (Hambrey and Dowdeswell, 1994)*." However, we are not sure we fully understand the reviewers suggestion about stating that the "ice flowlines are the surface manifestation of longitudinal foliation", as to us, 'ice flowlines' are simply the directions along which ice flows across the ice shelf, not only at the surface, but throughout the depth of ice.

Line 104: Please consider also citing Langley et al. (2016) and Arthur et al. (2020b) here, which record observations of down-ice lake advection on Langhovde Glacier and Shackleton Ice Shelf, East Antarctica.

We have done this.

Line 111: I suggest continuing the final sentence with: 'as described above, enabling it to support a large surface area of surface meltwater (Alley et al., 2018).

Thank for this good suggestion. We have added this text and reference.

Line 117: Be consistent with the use of northern/north GVIIS; I think northern is used most frequently throughout.

Previously we had used 'on north GVIIS' when trying to be brief (e.g. in sub headings, and figure captions), whereas we had used 'on the northern GVIIS' in the main paper text. However, we now use 'northern' (not 'north') throughout our revised paper, including in our title.

Line 125: What is the uncertainty associated with this microwave brightness temperature product and the ASCAT product in Section 3.2? There could be more discussion of this.

Regarding uncertainties with the microwave radiometer and scatterometer products, please see our response to this reviewer's general comment on pages 2 and 3 of this response letter.

Line 129: I suggest adding one sentence after this along the lines of: the sensor measures the emitted energy from the surface and sub-surface, proportional to the brightness temperature (which increases with the presence of liquid water and which increases absorption and emissivity).

We thank the reviewer for this helpful suggestion, though we have modified their suggested sentence slightly to avoid overlap with the previous sentence. We have insert the following two sentences at the beginning of the paragraph (lines 183 – 185): "*Microwave radiometer observations of melt, expressed as brightness temperatures, depend primarily on the snow temperature profile and emissivity (Zwally, 1977). When liquid water exists in the snow, there is a significant increase in the absorption, and therefore an increase in the microwave emissivity, resulting in higher brightness temperature.*"

Line 142: Briefly state why grid cells > 1700 m a.s.l. were masked out, presumably to only show data corresponding to ice shelf areas.

The reviewer is correct with their assumption. Cells > 1700 m a.s.l. were masked out such that only melt on ice shelves was predominantly analyzed. We have written this sentence to read as follows: "*Grid cells with surface elevations > 1700 m a.s.l. were masked out so that melt over the ice shelves was predominantly analysed and so that large topographic features (i.e., mountain peaks) in the radiometers field-of-view were avoided.*"

Line 148: Consider adding a citation here?

We have added a reference to Zwally (1977) here.

Line 161: I suggest adding a sentence explaining where this threshold comes from i.e. was based on empirical comparisons with QuikSCAT-derived melt (Ashcraft and Long, 2006).

We have added the following text to the end of this sentence: "*…..ASCAT signal is lower than the winter mean signal minus 3 dB, as proposed by Ashcraft and Long (2006) using a melt model and QuikSCAT Ku-band (13.4 GHz) observations*".

Line 166: Add 'scatterometer-derived' before 'cumulative melt days'.

We have done this.

Line 177: Perhaps briefly outline here the threshold-based algorithms other than NDWI that this method uses, i.e. NDSI and others, or else list which bands are used.

Adding details of all the bands and band combinations that we used to the manuscript would be extremely lengthy (at a minimum this would be two additional paragraphs, i.e. one paragraph per sensor, with 5 – 6 sentences in each paragraph), and given all this detail is in Moussavi et al (2020), which we follow exactly, we not think repeating this information in our paper is of value. We have, however, added a little more detail to the current text. This section now reads: "*Moussavi et al's (2020) method, developed separately for Landsat 8 and Sentinel-2, combines separate threshold-based algorithms to detect (1) lakes, (2) rocks, and (3) clouds. Optimal thresholds for each band and band combination (e.g. Normalized Difference Water Index (NDWI), Normalized Difference Snow Index (NDSI) and others) were determined by creating a training dataset based on selected Landsat 8 and Sentinel-2 images, which represented spectral properties of several classes (e.g. lakes, slush, snow, clouds, rocks, cloud shadows). Most notably, to classify liquid water-covered pixels, the NDWI is used*

*(Pope et al., 2016; Bell et al., 2017), with NDWI thresholds of > 0.19 and > 0.18 for Landsat 8 and Sentinel-2, respectively".*

Line 182: Consider also citing Bell et al. (2017) and Arthur et al. (2020b) here.

We have added these references.

Line 184: This depth-reflectance algorithm makes a number of assumptions which I think are worth briefly outlining in an additional sentence, including the assumption of homogenous lake bottom albedos, minimal wind-driven light scattering, etc (Sneed and Hamilton, 2007).

Please see our earlier response to a similar comment made by this reviewer on page 4 of this letter.

Line 185: There is no discussion currently in this section of false positives. How did you deal with these (if there were any), and were they manually removed?

The major issue with any image classification method, including ours, is false positives, which results in overestimation of lake areas. The only way to reduce these errors would be to post-process products, i.e. manual inspection of results, which is very labor intensive and is rather subjective. Therefore, we did not manually post process the images used in our study.

Line 218: I think 'warm' here more accurately describes foehn winds than 'hot'. Also suggest adding at the end of this sentence 'and commonly occur on the AP', and citing Luckman et al. (2014).

We agree re. foehn winds being 'warm' (not hot) and have changed this. We have also added these additional words and reference (line 330).

Line 219: I suggest adding an indication here of how steep the topography is, e.g. maximum slope.

Having shown that foehn winds are not a dominant contributor to surface melt during this 2019/20 season, we think that the further examination of drivers for foehn winds (including the slope of Alexander Island on the NW margin of the GVIIS) to be beyond the scope of our current study.

Line 339: This is an interesting finding. I wonder whether it would be worth adding an additional sentence (either here or at the end of this section) explicitly summarising these observations demonstrate you can still get record melt when conditions are generally warmer, with no involvement of foehn wind events.

This is a useful point, and we have added the following sentence to the Conclusions to explain this: "*It is therefore notable that although the high melt event over the northern GVIIS is 2019/2020 was caused by warmer than average air temperatures, such local weather conditions were not foehn-driven*". Also, we will add additional sentences to the Abstract and Results to emphasize that the high melt in 2019/2020 was not foehn driven.

Line 393: Are there any measurements of firn air content/thickness on George VI?

There are a couple of studies that have modelled firn air content (FAC) over GVIIS (e.g Ligtenburg et al., 2014), but the spatial resolution of those studies is insufficient for us to be able to state differences in FAC between the north and south GVIIS.

Line 410: I don't think you can necessarily suggest that surface melt volumes were highest in 2019/2020 out of the prior 31 seasons without having explicitly derived volume estimates from imagery pre-2013.

We thank the reviewer for their comment about this, and having thought about this paragraph some more, we have decided to delete this paragraph entirely.

Line 419: Change 'zero re-freeze' to 'no refreezing occurred'.

We have changed this.

Line 434: Explain 'shoulder seasons' – do you mean colder seasons (autumn/winter?).

By "shoulder seasons" we meant all seasons excluding summer, but mainly winter. However, we now realize that our terminology is not clear, and we apologize for that. We have reworded this sentence: "*This is predictable foehn flow behaviour, e.g., over the Larsen C, foehn winds are strongest in winter, when wind speeds are generally higher (Datta et al., 2019; Wiesenneker et al., 2018).*"

Line 446: Change 'back to 2013' to 'from 2013 to 2020' and suggest adding to the last part of the sentence: 'was also exceptional in areal extent and volume [..]'.

We have made these changes.

Line 454: Add AOI area here in brackets to remind readers of its size in comparison with Larsen B in the next sentence. Also, what was the maximum lake depth?

This is a good suggestion and we have added the area of the AOI (7850 km$^2$) in brackets. However we are not convinced that stating the maximum depth here in the Conclusions is suitable given this value appears to be rather an outlier.

Line 458: I suggest also citing Alley et al. (2018) here.

We have added this reference.

Line 460: Consider showing in a Supplementary Figure the lakes mapped in this study overlaid onto the areas classed as vulnerable to hydrofracture by Lai et al. (2020).

We thank the reviewer for this suggestion, but after careful consideration, we have not done this as we think that such a figure would be too 'busy' to be easily interpretable, and therefore would not serve it's purpose well. We also considered simply inserting just a map of the vulnerable areas of the GVIIS from Lai et al. (2020), but in which case, it makes more sense to simply refer the readers directly to the figure in that study (as we do).

Figure S5: I notice there are two particularly deep lakes – why, out of interest, do you suggest this is?

We think that these areas may remain as topographic depressions from season to season due to the flow regime in this area. Additionally, their basin depths may have been enhanced due to enhanced lake bottom ablation in these areas. In other words, the greater the number of days that meltwater ponds in a specific area, the greater the effect of enhanced melting at the lake bottom due to the lower

albedo of the lake water compared to the surrounding water. However, as our study focusses on the total volumes of surface meltwater each season, rather than behaviours of specific lakes, we think that it is beyond the scope of the current study to discuss the evolution of these deep lakes in the paper.

Line 31: minor point, but check here and throughout that citations are listed chronologically.

We have done this.

Line 49: Add comma after 'However'.

We have done this.

Line 100: 'most extensive' rather than 'the largest'?

We have changed this.

Line 106: Change en-echelon to 'en-echélon' and consider adding a brief description for those unfamiliar with this term, e.g. closely spaced, sub-parallel.

Thank you, we have made both of these changes.

Line 120: Change '(from 2013)' to 'from 2013-2020' and the same with '(from 2017)'.

We have done this.

Line 125: Insert 'passive' before 'microwave radiometer'.

As a radiometer is, by definition, a 'passive' sensor, we have instead reworded this part of the sentence to read "… *was derived from microwave radiometer (i.e. passive) observations…".*

Line 132: Hyphenate 25 km.

We have done this.

Line 157: Hyphenate 4.45 km and consider briefly outlining the SIR algorithm.

We have reworded this sentence to read (lines 229 – 230): "*The 4.45-km enhanced product is obtained by applying the Scatterometer Image Reconstruction (SIR) algorithm with filtering (Lindsley and Long, 2016), which is used to improve the spatial resolution of irregularly and oversampled data (Early and Long, 2001).*"

Line 159: Hyphenate 'SMMR-based'.

We have done this.

Line 173: Rephrase to 'to selected multispectral imagery (see paragraph below)'.

We have done this.

Line 197: Are italics needed in this paragraph?

We have removed these italics.

Line 216: Remove 'periods'.

We have done this.

Line 220: citation should be Wiesenneker et al., 2018 (full reference is correct).

We have corrected this (however we note that the correct spelling is "Wiesenekker", not Wiesenneker).

Line 291: remove commas after 2020 and AOI.

We have done this.

Line 327: Change 'refreeze' to 'refreezing'

We have done this.

Line 390: Re-word sentence to 'meltwater ponding is not observed in the optical imagery until mid-December'.

We have done this.

Line 495: Please add volume, issue and page numbers: 44(6), 837-869.

We have done this.

**Reviewer #2**

This is an engaging study investigating the 2019/2020 austral melt season on the north George VI Ice Shelf, Antarctic Peninsula. The authors combine different datasets such as satellite microwave radiometer/scatterometer data, optical satellite imagery and local meteorological station data. The authors highlighted exceptional melt and surface ponding conditions in 2019/2020, and they concluded that warm temperatures were likely triggered by warm northwesterly and northeasterly low-speed winds, rather than Foehn winds.

The paper is well written, and the results are supported by the data and methods. The study has the potential to become a fully relevant article to concept of The Cryosphere and my criticisms are with regards presentation rather than science.

We thank this reviewer for their very useful remarks.

Specific comments:

Lines 31-32: Could you please specify the rate of mass loss?

*We will have reworded this sentence (line 34): "The rate of mass loss from the AP has tripled since the 1990s, with an average of 24 Gt yr-1 from 1979 to 2017 and an acceleration of 16 Gt yr[1] per decade (Rignot et al., 2019)." We also note, that as these statistics were noted in the Section 2 of our paper ("Study Site"), we have deleted them from that section in our revised paper.*

Lines 53-54: Is there any projected estimation of contribution of the AP ice shelf melting to the global sea level rise? If yes, that information would be useful.

*We have added detail about the potential sea level contribution from the George VI versus the Larsen C ice shelves at the end of Section 2 ("Study Site"). The new sentences are as follows: "Due to the strong buttressing forces that the GVIIS provides relative to the large volume of grounded ice in Palmer Land, if this ice shelf were to completely collapse, the resultant acceleration of the inland glaciers would add < 8 mm to global sea levels by 2100 and < 22 mm by 2300 (Schannwell et al., 2018). In contrast, sea level contributions resulting from the collapse of the much larger Larsen C Ice Shelf would be relatively low (< 2.5 mm by 2100, < 4.2 mm by 2300)."*

Lines 56-58: I would add a link of official WMO status on this evaluation: https://public.wmo.int/en/media/news/new-record-antarctic-continent-reported. In addition, to my knowledge, WMO is evaluation the value of 18.4∘C., not the 20.75∘C. There has been no official effort/information to evaluate the value of 20.75∘C. Further- more, this value was not recorded at the official Marambio station. I recommend the authors to make sure that whether this temperature is being evaluated or not. If not, I recommend them to remove the statement in lines 58-60.

*Thank you, we have added this weblink. We also agree that this 20.75°C record does not now appear to be in evaluation by the WMO, so we have deleted this sentence (in addition to removing the following sentence about the prior record on Signy Island, which is also no longer relevant).*

Lines 60-63: Could you please quantify melt amount in this part too? Because, you are comparing GVIIS with the other AP ice shelves. Some numerical values would help the reader to see the differences.

*Quantifying melt volumes on ice shelves in addition to the GVIIS is far beyond the scope of this study, however we have added an additional sentence to this section that qualitatively describes meltwater ponding on AP ice shelves that is observed in the MODIS mosaic shown in Figure 1a: "However, as Fig. 1a shows, in 2019/2020, surface meltwater ponding was also prevalent on the northwestern Larsen C (Bevan et al., 2020), the eastern Wilkins (also visible in the bottom left corner of Fig 1b), and the northern and northwestern Bach ice shelves."*

Lines 87-94: What makes the northern GVIIS more vulnerable to high surface summer melt rates? I would expect to consider rates of basal melting, nonetheless, you state that rates of basal melting are greatest at the southern end of the GVIIS. Therefore, a clarification might be needed to have a better idea on different physical processes at the northern and southern ends.

*The northern GVIIS experiences warmer air than the southern GVIIS (e.g. see Datta et al., 2018). Additionally, precipitation rates in the North are lower than in the south, which is at least partly attributable to the precipitation shadow of Alexander Island. We have rewritten the first sentence of this paragraph to read: "Compared to the southern GVIIS (~72°00' S to 77°00), the northern GVIIS (~70°30' S to ~72°00' S) experiences higher surface summer melt rates (< 250 mm w.e. yr-1; Trusel et al., 2013; Datta et al., 2018) and lower accumulation rates (< 0.2 Mg m-2 yr-1; Bishop and Walton, 1981;*

*Reynolds, 1981); the latter is attributed to the presence of a precipitation shadow down-wind of Alexander Island (Bishop and Walton, 1981)."*

Regarding basal melting, we are not sure what this reviewer means by "I would expect to consider rates of basal melting", as we do already give detailed information about the spatial variation in basal melt rates, and the reasons for high basal melt rates, in the current version of our paper (current lines 91 – 94). However in our revised paper, we also state basal melt rates from Adusumilli et al. (2020), as requested by Reviewer 1.

Lines 123, 208-212: Is there any elevation difference between the AWS station and ice shelf?

This is a good question. The AWS elevation is 66 m, which is similar to the ice shelf surface elevation in this region (50 – 60 m, given that the ice shelf thickness is 500 – 600 m). We now state the AWS's elevation along the AWS's coordinates in Section 3.4.

Line 213: What does it mean exactly "using 12-hour data" for the 1999-2020 period? I understand that you use only one time step as a daily mean temperature for the 1979- 1999 period. If yes, what is the exact hour of the observation (morning, late afternoon)?

We apologize for being unclear. By "12-hour data", we are referring to data that were recorded 12 hours apart; at noon and midnight. Therefore each daily mean value from 1979 to 2020 (NB. not just 1999 to 2020 as this reviewer states) is calculated from two data points. We have added these details to the manuscript.

For the second period, did you pick up the same hour with the 1999-2020 period or did you take a temporal mean of 12-hour data?

We perform high temporal resolution analysis from 2007 to 2020, which is (as it currently stated in the manuscript) when "*when AWS data are at a high frequency (10 minute intervals) and data gaps are minimal*". Therefore, for this period, we use the 10 minute data (not the 12-hour data) to calculate the length of time when temperatures are continuously at/above $0^{o}C$. We have clarified this in our paper.

Line 223: Could you please specify why there exist lower mean speeds over north GVIIS?

Although the study by van Wessem et al (2015; their Figs 10 and 11) show that in general, wind speeds over North GVIIS appear to be similar to those in the Scar Inlet/Larsen C region, foehn winds will likely be slower in speed at Fossil Bluff compared to the Cabinet Inlet due to the lower mean elevation of Alexander Island's topography compared to the AP mountains to the west of the Scar Inlet (e.g. see van Wessem et al (2015); their Fig 1). Therefore, we have written this sentence to read: "*We use a wind speed threshold of 1.5 ms$^{-1}$ instead of the higher threshold of 3.5 ms$^{-1}$ used by Datta et al. (2019) for the Cabinet Inlet AWS to account for lower foehn wind speeds over north GVIIS, which result from the lower mean elevation of the mountains on Alexander Island compared to those on the AP west of Cabinet Inlet (van Wessem et a., 2015).*"

We also note that our wind speed threshold of 1.5 ms$^{-1}$ is actually more liberal than the 3.5 ms$^{-1}$ threshold used by Datta et al (2019). In other words, although we only calculated 9 hours of foehn conditions in the 2019/2020 melt season, if we had used the higher Datta et al. threshold, we would have calculated even fewer hours of foehn conditions for the northern GVIIS.

Lines 238-239: In your analysis, I understand that Larsen C shows a record high melt year in 1992-1993, not in 2019-2020 on the contrary to the findings of Bevan et al. Is that related just with the different datasets? Can you please specify potential uncertainties of each product?

We have added the following sentence to our revised manuscript, which we hope more clearly describes the discrepancy (lines 370 – 376): "*This finding is contrary to the results of Bevan et al. (2020), who report that Larsen C experienced a 41-year record high melt year in 2019/2020. Bevan et al's (2020) results are based on microwave radiometer (SMMR/SSMI) data for melt seasons from 1979/1980 until 2016/2017, followed by microwave scatterometer (ASCAT) data from 2017/2018 to 2019/2020. In contrast, we use SSMR/SSMI data over the AP for the full 1979 to 2020 period to preserve consistency. As we explain in Section 3.2, ASCAT C-band radar is likely to be more sensitive to melt at depth than microwave radiometers, thus resulting in Bevan et al's (2020) higher calculated melt over Larsen C in the 2019/2020 season after combining different data sources into one time series.*"

Regarding the potential uncertainties of each of the melt products, this has now been addressed in response to one of Reviewer 1's main comments, so please refer to our reply on pages 2 and 3 of this document.

Lines 326-334: Why not to compare the longest warm periods with the volume changes? For instance, are the longest periods coincide with the largest volume change?

We thank the reviewer for this suggestion, and we have inserted the following sentence into the Results section (4.3): "*In 2019/2020, volumes of meltwater ponding are highest in early January and then again in early February; corresponding with periods when air temperatures are ≥ 0°C for extended periods.*"

Additionally, we have inserted the following sentence into the Discussion section (i.e. '5.3. Local climatic controls of the 2019/2020 melt event'): "*For example, we note that sustained warm temperatures in early January (Fig. 7b, periods A and B) and early February (Fig. 7b, periods C and D) coincide with periods when surface meltwater volumes derived from optical imagery are relatively high (Fig. 5b).*"

Lines 335-339: I think the authors should also discuss the potential role of the warm air advection given the low foehn conditions. I suspect that regional sensible heat transport seems to be one of the main contributors of the 2019/2020 melt season, particularly for the first two weeks of February.

We completely agree that warm air advection likely drove the sustained high air temperatures in 2019/2020, and therefore the high melt in this season. However, we are wondering if the reviewer wrote this comment before reading the following two paragraphs in our paper, which describe the results of our analysis of wind speed alongside air temperature data (i.e. warm air advection). In particular, our our paper states: "*in the 2019/2020 melt season, we show that both northwesterly and northeasterly winds show warmer temperatures at lower wind speeds*" in Results section 4.4. Melting due to the increase of warm, low-speed winds from the NE and NW is also discussed in the Discussion, and mentioned in the Conclusion and Abstract.

To make it 100% clear that we are indeed describing warm air advection, in our revised manuscript, we will add the following sentence to the beginning of the fourth paragraph in Section 4.4: "*Finally, we analyse the potential role of warm air advection, resulting in sensible heat transport, on the high melt in 2019/2020.*"

Lines 349-359: Can you specify where the calculated 9 hour of foehn is located in this time series (Fig. 6)?

*The time periods when foehn conditions are present are (and were previously) described in the previous paragraph that specifically focusses on foehn winds, and these periods are (and were) labelled in Fig 6b (blue circles), so we are wondering if the reviewer missed this. The relevant sentence currently reads: "Foehn conditions (as described in Section 3.4) are only present for about 9 hours over the entire 2019/2020 season (Fig 5, blue line), and occur in early and late summer (Fig. 6b, blue circles) when winds are typically stronger."*

Lines 431-438: I am a bit disconnected here. So, I understand that rather than local conditions (i.e., Foehn winds) one should expect to see regional- and/or large-scale warm advection. In this case, I would expect to see similar warming rates over the GVIIS and Larsen C. However, in Fig. 2c there are notable warming differences be- tween these two regions. Do the authors have any idea for these differences?

*We would first like to clarify to this reviewer that Figure 2 shows surface melt, not warming (though surface melt is obviously driven by the surface energy balance). Second, we agree that this figure shows significantly higher melt over the George VI and Wilkins ice shelves compared to the NE areas of the AP (inc. Larsen C) in 2019/2020, and we have added the following sentence to Results section 4.1 to make this fact clearer: "For the AP, cumulative melt days in the 2019/2020 austral melt season are highest in the southwestern areas of the AP (including the Wilkins and George VI ice shelves), in addition to the northern area of the Larsen C Ice Shelf (Fig. 2b). In comparison, cumulative melt days in 2019/2020 are relatively low over the southern areas of the Larsen C." However, as the main aim of this study is to focus on melt on the northern GVIIS, and the local atmospheric controls of this melt, it is outside of the present study's scope to perform modelling to establish why the SW area of the AP experienced more melt than the NW area of the AP in 2019/2020.*

Fig. 1 and Fig. 2: Could you please use a larger font size for the lat/lon coordinates?

*We have done as suggested for Figs 1 and 2, as well as for Fig S1.*

Overall suggestion: As you use different datasets with different spatial and temporal resolutions, I would add a table for the information given in data and methods (e.g., name of the product, resolution, temporal coverage etc.). This makes it easier for reader to follow the result.

*As we actually only use four data products in total, i.e. two microwave products (SMMR/SSMI and ASCAT) and two optical image products (Landsat 8 and Sentinel-2), we do not think that a table of details about these products will be sufficiently beneficial, especially as the details about each product are clearly stated at the beginning of each section in the Methods (i.e. SMMR/SSMI is described in Section 3.1, ASCAT is described in Section 3.2, and Landsat 8/Sentinel-2 are described in Section 3.3). The Editor has confirmed that they agree with our decision on this.*

**Reviewer #3**

This is a well-written and evidenced paper that uses several independent datasets to explore the precise characteristics melting over the George VI Ice shelf. The authors demonstrate that melt extent and duration over GVIIS was higher in 2019/20 than in any other melt year analysed, and suggest that

localised meteorological factors were responsible. Namely, persistent north-westerly and north-easterly low-speed winds, which allowed temperatures to rise above freezing and for melt to occur.

The manuscript is presented in a logical structure, with interesting and informative figures. The methods used are justified and largely given in sufficient detail to be reproducible. The results are supported by the data presented and the conclusions drawn are consistent with the evidence given. I recommend the publication of this manuscript in TC subject to minor revisions.

We thank this reviewer for their positive remarks about our study and we are very grateful for the useful comments that they have provided.

Specific comments:

[L49-52] You make reference to the collapse of Larsen A (and Larsen B in a previous sentence) and then in the next sentence to the partial collapse of Larsen A and B – could you revise this to be clearer/more consistent with the above?

We have made these sentences clearer by initially talking about the near-complete collapse events of the Larsen A (and Larsen B), before then talking about the partial collapse events of these (and other) ice shelves.

[L57-61] I think the 20.75 record has been thrown out by the WMO so you can probably remove this part of the sentence. The justification can be made using the Esperanza record alone.

Thank you for highlighting this, which was also noted by Reviewer #2. We have removeed all reference to the potential 20.75C record.

[L62] You could link this back to the Bevan et al. paper again.

As we do not have confidence in the results of the Bevan et al. paper (i.e. for the reasons discussed in Section 4.1 of our paper), we would rather not add an additional reference to this paper.

[Section 3.4] Can you comment on the data processing for the AWS – e.g. is it an instantaneous value every 12 hours, or an average?

A similar question was asked by Reviewer #2, so we apologize for not being clear. The 12-hour data include instantaneous temperature measurements at noon and midnight, so we calculate the mean of those two data points to calculate the daily mean temperatures. We have clarified this is our revised manuscript (line 323).

[L172] extra "to" after parenthesis

We have corrected this.

[para starting 187] Is there precedent for this type of methodology?

This methodology follows Moussavi et al. (2020), which is clearly stated in our revised manuscript.

[para starting L320] It may be worth commenting on the fact that surface temperatures can differ from 2 m air temperatures and so 0 degrees at 2 m may not actually mean surface temperatures are at the melting point / vice versa. How might this influence the length of periods of >0 temperatures and consequently refreezing?

We thank the reviewer for this good suggestion, and will have added the following sentence to the Methods (section 3.4): "*Although the air temperature measured at a height of 2-m by the AWS will vary slightly to that at the ice surface, for the purposes of this study, we assume these temperatures to be equivalent (Kuipers Munneke et al., 2012)*"

Additionally, we would like to state that we did experiment with alternative thresholds of -2°C and -1°C (accounting for the possibility of melt occurring below 0°C), and the overall result was unchanged; 2019/2020 still experienced the longest periods of high air temperatures and thus we elected to use the simplest threshold of 0°C.

[para starting L335] During how much of the time series in Fig. 4a was the temperature above 1 standard deviation above the mean? This might be another interesting way to think about this anomalously warm period.

This is certainly true and we have added the following sentence to this section to clarify this: "*We also note that the temperature during these five periods is often more than one standard deviation greater than the multi-year daily mean*" (lines 533 – 535).

[para starting 335] Was any sensitivity testing conducted regarding the foehn detection method? Different thresholds and methods can yield very different results, e.g. isentrope-based method of King et al. (2017, doi: 10.1002/2017JD02680) vs surface method similar to the one used here. Varying the thresholds used can also make a   difference (e.g. Turton et al., 2018, doi.org/10.1002/qj.3284).

Sensitivity testing regarding the wind threshold in the foehn condition calculations was conducted. We note that by decreasing the windspeed threshold, the sensitivity of the foehn-detection algorithm increases substantially. Therefore, we choose a liberal threshold of 1.5 ms$^{-1}$, and with that threshold, we calculated just 9 hours of foehn conditions in the 2019/2020 melt season. With the 3.5 ms$^{-1}$ threshold used by Datta et al (2019), there were even fewer hours of foehn conditions calculated in this melt season. To further refine this wind threshold for the GVIIS, a more in-depth study, using additional AWS data from elsewhere on the ice shelf (e.g. following the approach of Turton et al. 2018), potentially in combination with high-resolution modeling (e.g. following the approach of King et al. 2017), would be required.

Additional references that we have added

Alley KE, Scambos TA, Anderson RS, et al. Quantifying vulnerability of Antarctic ice shelves to hydrofracture using microwave scattering properties. Remote Sensing of Environment 210: 297–306, https://doi.org/10.1016/j.rse.2018.03.025, 2018.

Arthur, J. F., Stokes, C. R., Jamieson, S. S. R., Carr, J. R., and Leeson, A. A.: Distribution and seasonal evolution of supraglacial lakes on Shackleton Ice Shelf, East Antarctica, The Cryosphere Discuss., https://doi.org/10.5194/tc-2020-101, 2020b.

Bishop, J. F. and Walton J. L. W. Bottom melting under George VI Ice Shelf, Antarctica. Journal of Glaciology. 27(97), 429-447, 1981

Datta, R. T., Tedesco, M., Agosta, C., Fettweis, X., Kuipers Munneke, P., & van den Broeke, M. R. Melting over the northeast Antarctic peninsula (1999–2009): Evaluation of a high-resolution regional climate model. The Cryosphere, 12(9), 2901–2922. https://doi. org/10.5194/tc-12-2901-2018, 2018.

Kuipers Munneke, P., van den Broeke, M. R., King, J. C., Gray, T., and Reijmer, C. H.: Near-surface climate and surface energy budget of Larsen C ice shelf, Antarctic Peninsula, The Cryosphere, 6, 353–363, https://doi.org/10.5194/tc-6-353-2012, 2012.

Langley ES, Leeson A. A., Stokes. C. R., et al. Seasonal evolution of supraglacial lakes on an East Antarctic outlet glacier. Geophysical Research Letters 43(16): 8563–8571, 2016.

Lucchitta, B. K. and Rosanova, C. E. Retreat of northern margins of George VI and Wilkins Ice Shelves, Antarctic Peninsula. Annals of Glaciology 27: 41–46, 1998.

Luckman A, Elvidge A, Jansen D, et al. Surface melt and ponding on Larsen C Ice Shelf and the impact of fohn winds. Antarctic Science 26(6): 625–635, 2014.

Schannwell, C., Cornford, S., Pollard, D., and Barrand, N. E.: Dynamic response of Antarctic Peninsula Ice Sheet to potential collapse of Larsen C and George VI ice shelves, The Cryosphere, 12, 2307–2326, https://doi.org/10.5194/tc-12-2307-2018, 2018.

Trusel, L. D., K. E. Frey, S. B. Das, P. Kuipers Munneke, and M. R. van den Broeke, Satellite-based estimates of Antarctic surface meltwater fluxes, Geophys. Res. Lett., 40, 6148–6153, doi:10.1002/2013GL058138, 2013.